# Anchor-guided Hypergraph Condensation with Dual-level Discrimination

**Fan Li** [1]  **Xiaoyang Wang** [1]  **Chen Chen** [2]  **Wenjie Zhang** [1]

## Abstract

The increasing prevalence of large-scale hypergraphs poses significant computational challenges for hypergraph neural network (HNN) training. To address this, hypergraph condensation (HGC) distills large real hypergraphs into compact yet informative synthetic ones, beyond graph condensation (GC) methods limited to pairwise relations. However, existing HGC methods rely on decoupled training architectures, where structure generators are pre-trained on the original hypergraph but not jointly optimized with condensed features during refinement, resulting in misaligned structures that degrade downstream utility. Moreover, trajectory-based optimization incurs substantial computational overhead in refinement, limiting condensation efficiency. To tackle these issues, we propose **A**nchor-guided **H**yper**G**raph **C**ondensation with **D**ual-level **D**iscrimination (**AHGCDD**), which consists of three key components: (1) a node initialization module based on Heat Kernel PageRank (HKPR) to encode structural knowledge into feature semantics; (2) an anchor-guided hyperedge synthesis strategy for joint optimization of condensed features and structure; (3) a theoretically grounded dual-level discrimination objective for utility-preserving condensation without redundant HNN training. Extensive experiments demonstrate the superior effectiveness and efficiency of AHGCDD.

## 1. Introduction

Hypergraphs provide a powerful tool for modeling higher-order interactions among multiple entities and have been successfully applied across diverse domains, including social analysis (Do et al., 2020), biochemistry (Yang et al., 2023b), and e-commerce (Han et al., 2023). Hypergraph Neural Networks (HNNs) (Yadati et al., 2019; Chien et al., 2022; Saxena et al., 2024) have achieved remarkable progress in hypergraph representation learning, owing to their strong capability to capture and leverage higher-order information. However, the increasing prevalence of large-scale hypergraph data incurs prohibitive computational costs for HNN training (Kim et al., 2023; Tang et al., 2025), which not only hinders practical deployment but also restricts the exploration of other promising directions in hypergraph learning, such as neural architecture search (Lin et al., 2024; Antelmi et al., 2023) and continual learning (Fu et al., 2023).

To address the challenges brought by the scale of graph data, graph condensation (GC) has emerged as a promising data-centric solution, due to its high compression ratio and lossless performance (Liu et al., 2024b; Gao et al., 2025b). It aims to synthesize and optimize a compact graph such that Graph Neural Networks (GNNs) (Kipf & Welling, 2017; Hamilton et al., 2017) trained on it achieve performance comparable to those trained on the original full graph. To this end, existing GC methods typically optimize synthetic graphs by matching surrogate objectives with those of the original data, such as gradients (Jin et al., 2022; Yang et al., 2023a), training trajectories (Zheng et al., 2023; Zhang et al., 2024), and embeddings (Liu et al., 2022; Xiao et al., 2024). By incorporating essential training knowledge from the original graph into the condensed data, GC facilitates efficient GNN training with only minor performance loss.

Despite the considerable progress achieved by graph condensation, extending these methods to hypergraphs remains non-trivial due to the need to model high-order interactions and handle the exponential complexity of hyperedges. To bridge this gap, HG-Cond (Gong et al., 2025) introduces the first hypergraph condensation framework with a decoupled training paradigm, consisting of Neural Hyperedge Linker (NHL) pre-training followed by an amelioration module. Specifically, the NHL is pre-trained via variational inference to capture high-order connectivity with linear complexity, while the amelioration module refines the hypergraph by matching multiple surrogate objectives. As a result, the condensed data substantially improves HNN training efficiency while maintaining strong effectiveness and generalization.

[1]School of Computer Science and Engineering, University of New South Wales, Sydney, Australia [2]School of Artificial Intelligence, Shenzhen University, Shenzhen, China. Correspondence to: Chen Chen <chen_chen@uow.edu.au>.

*Proceedings of the 43rd International Conference on Machine Learning*, Seoul, South Korea. PMLR 306, 2026. Copyright 2026 by the author(s).

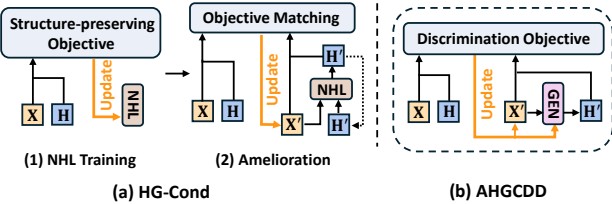

*Figure 1.* Learning pipelines of HG-Cond and AHGCDD.

However, this decoupled architecture with matching-based amelioration exhibits two key limitations in hypergraph condensation. (1) *Misaligned Structure Generation.* As shown in Figure 1(a), HG-Cond decouples the optimization of condensed node features and the structure generator, i.e., the NHL. Although the NHL is pre-trained to preserve structural information by reconstructing the original hypergraph, it remains fixed during the amelioration stage, where the optimization focuses on aligning the model's training behavior on the condensed graph with that on the original graph. This lack of joint optimization between the generated structure and condensed features leads to an objective misalignment, preventing the synthesized structure from effectively preserving training utility and ultimately degrading downstream performance. (2) *Resource-Intensive Optimization.* During amelioration, HG-Cond refines the synthetic hypergraph using a Gradient–Parameter Synergistic Matching (GPSM) objective, which requires repetitive HNN retraining, incurring substantial computational overhead. In addition, the variational pre-training of NHL introduces significant memory consumption. These factors make HG-Cond resource-intensive and difficult to scale to large hypergraphs.

To address the above issues, we propose AHGCDD, a novel hypergraph condensation framework via anchor-guided hyperedge generation and dual-level discrimination. Specifically, we first design an HKPR-based node initialization module to adaptively integrate high-order information from local to global neighborhoods into feature semantics, yielding enriched condensed node features. On top of this, we introduce an anchor-guided hyperedge synthesis scheme, where each node sequentially acts as an anchor to induce a hyperedge. Within each hyperedge, high-order interactions between the anchor and candidate nodes are synthesized based on feature-level distances parameterized by a learnable function. This design enables end-to-end optimization of the condensed structure and features, avoiding optimization misalignment while eliminating the need for costly pre-training of the structure generator. In addition, for each anchor-induced hyperedge, we introduce a learnable sparsity threshold, enabling flexible and adaptive control of hyperedge density. During optimization, we propose a dual-level discrimination loss to preserve the training utility of condensed data, consisting of coarse-grained class prototype discrimination and fine-grained sample-wise discrimination. A dynamic weighting mechanism is employed to effectively

balance the two loss terms during training, thereby improving convergence. Theoretical analysis further justifies that our objective preserves the global distribution of the original hypergraph while achieving refined inter-class separability. By optimizing with the discrimination loss, we eliminate the need for repetitive HNN training, significantly improving condensation efficiency. Our main contributions can be summarized as follows:

- We propose AHGCDD, a novel hypergraph condensation framework with an anchor-guided hyperedge generation scheme. To the best of our knowledge, it is the first unified architecture that jointly optimizes node features and high-order hypergraph structures.

- To generate structurally enriched condensed features, we design an HKPR-based node initialization module. On this basis, an anchor-guided hyperedge synthesis strategy is introduced to generate high-order interactions via feature-level associations, enabling joint optimization of condensed features and structures.

- We develop a dual-level discrimination loss to align training utility during optimization without reliance on HNN training. A comprehensive theoretical analysis justifies the effectiveness of the proposed objective.

- Extensive experiments demonstrate that AHGCDD outperforms state-of-the-art methods in effectiveness and generalization, while substantially improving condensation efficiency, achieving up to 144× speedup.

## 2. Preliminary

This section first introduces the notations used throughout this paper and then formalizes the hypergraph condensation problem. Background on hypergraph neural networks (HNNs) is provided in the Appendix B.1.

**Notations.** Let $\mathcal{T} = (\mathcal{V}, \mathcal{E}, \mathbf{X}, \mathbf{Y})$ denote a hypergraph, where $\mathcal{V} = \{v_i\}_{i=1}^{|\mathcal{V}|}$ is the node set and $\mathcal{E} = \{e_j\}_{j=1}^{|\mathcal{E}|}$ is the hyperedge set, with $|\mathcal{V}| = N$ and $|\mathcal{E}| = M$. $\mathbf{X} \in \mathbb{R}^{N \times d}$ is the $d$-dimensional node feature matrix. $\mathbf{Y} \in \mathbb{R}^{N \times C}$ denotes the one-hot label matrix over $C$ classes. The hypergraph structure is represented by an incidence matrix $\mathbf{H} \in \mathbb{R}^{N \times M}$, where each entry $h_{ij} = 1$ if $v_i \in e_j$ and $h_{ij} = 0$ otherwise. The diagonal matrices of node and hyperedge degrees are denoted by $\mathbf{D}_v \in \mathbb{R}^{N \times N}$ and $\mathbf{D}_e \in \mathbb{R}^{M \times M}$, respectively.

**Problem Definition.** Given a large-scale hypergraph $\mathcal{T} = (\mathbf{X}, \mathbf{H}, \mathbf{Y})$, the goal of hypergraph condensation is to learn a downsized synthetic hypergraph $\mathcal{S} = (\mathbf{X}', \mathbf{H}', \mathbf{Y}')$ with $\mathbf{X}' \in \mathbb{R}^{N' \times d}, \mathbf{H}' \in \mathbb{R}^{N' \times M'}$ and $\mathbf{Y}' \in \mathbb{R}^{N' \times C} (N' \ll N, M' \ll M)$, such that the HNN model trained on $\mathcal{S}$ can achieve comparable performance to that trained on the original large $\mathcal{T}$. Following (Zheng et al., 2023; Liu et al.,

2024a), $\mathbf{Y}'$ is pre-defined based on the class distribution of the original label space $\mathbf{Y}$. The condensation objective can be formulated as a bi-level optimization problem:

$$\min_{\mathcal{S}} \mathcal{L}(f_{\theta_{\mathcal{S}}}(\mathbf{H}, \mathbf{X}), \mathbf{Y})$$
$$\text{s.t.} \quad \theta_{\mathcal{S}} = \arg\min_{\theta} \mathcal{L}(f_{\theta}(\mathbf{H}', \mathbf{X}'), \mathbf{Y}'), \qquad (1)$$

where $\mathcal{L}$ denotes the loss function that measures the node classification error (e.g., cross-entropy), $f_{\theta}$ represents the HNN model parameterized by $\theta$, and $\theta_{\mathcal{S}}$ denotes the model parameters trained on the synthetic hypergraph.

## 3. The Proposed Method: AHGCDD

In this section, we present our proposed AHGCDD, which consists of three essential modules. An overview of the framework is shown in Figure 2. The first module, termed the HKPR-based node initialization, incorporates higher-order structural dependencies from local to global neighborhoods into the node features. The second module, anchor-guided hyperedge generation, leverages feature-level relationships to synthesize high-order interactions with adaptive sparsity control. Finally, a dual-level discrimination module jointly optimizes the condensed features and structure via coarse-grained prototype discrimination and fine-grained instance-level discrimination, while incurring only minor computational overhead.

### 3.1. HKPR-based Node Initialization

In the node initialization stage, we fuse high-order structural information with feature semantics into the condensed node features. By incorporating structural information at initialization, the enriched features serve as a more expressive starting point for condensed data optimization and lead to improved downstream utility. Moreover, these structure-aware features provide a solid foundation for the subsequent feature-driven hyperedge generation (Section 3.2) by supplying informative topological knowledge.

To adaptively capture multi-scale structural dependencies from local neighborhoods to the global context, we draw inspiration from Heat Kernel PageRank (HKPR) (Chung, 2007) and propose a novel hypergraph diffusion mechanism. This mechanism aggregates paths of varying lengths between nodes, with their importance governed by the Poisson distribution, as follows:

$$\tilde{\mathbf{X}} = \sum_{k=0}^{\infty} \frac{e^{-\lambda}\lambda^k}{k!} \cdot \mathbf{P}^{(k)}\mathbf{X},$$
$$\text{where} \quad \mathbf{P} = \mathbf{D}_v^{-\frac{1}{2}}\mathbf{H}\mathbf{D}_e^{-1}\mathbf{H}^{\mathbf{T}}\mathbf{D}_v^{-\frac{1}{2}}, \qquad (2)$$

where $\lambda \in \mathbf{N}^+$ is the parameter that controls the path importance. The following theorem indicates that HKPR-based

diffusion performs exponential low-pass filtering on the hypergraph spectrum, suppressing high-frequency noise and promoting smooth structural information propagation. For brevity, we denote $\mathbf{P}_{\lambda} = \sum_{k=0}^{\infty} \frac{e^{-\lambda}\lambda^k}{k!} \cdot \mathbf{P}^{(k)}$.

**Theorem 3.1.** *Let $\mathcal{L} = \mathbf{I} - \mathbf{P} \in \mathbb{R}^{n \times n}$ denote the normalized hypergraph Laplacian, with eigenvalues $0 = \mu_1 \leq \mu_2 \leq \cdots \leq \mu_n \leq 2$. The HKPR-based diffusion $\tilde{\mathbf{X}} = \mathbf{P}_{\lambda}\mathbf{X}$ can be expressed as a spectral filtering operation in the hypergraph Fourier domain:*

$$\tilde{\mathbf{X}} = \mathbf{U}\, g(\mathbf{M})\, \mathbf{U}^{\top}\mathbf{X}, \qquad (3)$$

*where $\mathbf{U} = [\mathbf{u}_1, \ldots, \mathbf{u}_n]$ is the orthonormal eigenbasis of $\mathcal{L}$, $\mathbf{M} = \mathrm{diag}(\mu_1, \ldots, \mu_n)$ is the Laplacian spectrum, and $g(\mu) = e^{-\lambda\mu}$ is an exponentially decaying low-pass filter.*

The proof of Theorem 3.1 can be found in Appendix E.1. However, directly computing Eq. (2) is intractable, as it entails an infinite-order diffusion process. In practice, we employ a truncated approximation with a finite order $K$ as:

$$\tilde{\mathbf{X}} \approx \sum_{k=0}^{K} \frac{e^{-\lambda}\lambda^k}{k!} \cdot \mathbf{P}^{(k)}\mathbf{X}. \qquad (4)$$

We next discuss the principle for choosing the truncation order $K$. Note that the diffusion step in HKPR follows a Poisson distribution, and our truncated approximation aims to retain contributions from high-probability step lengths while discarding the tail event. The following lemma provides an upper bound on the Poisson tail probability.

**Lemma 3.2.** *Let $N \sim \mathrm{Poisson}(\lambda)$ with parameter $\lambda > 0$. Then for any $t > 0$, we have*

$$\mathbb{P}\left[N \geq \lambda + t\sqrt{\lambda}\right] \leq \exp\left(-\frac{t^2}{2 + t/\sqrt{\lambda}}\right). \qquad (5)$$

The proof of Lemma 3.2 is provided in Appendix E.2. To achieve a better trade-off between approximation accuracy and computational efficiency, we set $K = \lceil \lambda + 3\sqrt{\lambda} \rceil$ in this work. Under this setting, the tail probability beyond $K$ decays exponentially and is negligible, ensuring that the truncated diffusion captures the dominant diffusion steps while controlling the computational cost.

Given the $N$ structure-aware features $\tilde{\mathbf{X}}$, we map them to $N'$ initialized nodes following (Li et al., 2025b). Specifically, for each synthetic node $v_i'$ in class $c$, we uniformly sample a subset $\mathcal{S}_i$ of original nodes from the same class and aggregate their features via mean pooling:

$$\mathbf{X}_i' = \frac{1}{|\mathcal{S}_i|}\sum_{j \in \mathcal{S}_i} \tilde{\mathbf{X}}_j, \quad \mathcal{S}_i \subseteq \mathcal{I}_c, \qquad (6)$$

where $\mathcal{I}_c = \{i \mid y_i = c\}$ denotes the indices of original nodes in class $c$, and we fix the number of sampled nodes per synthetic node to $|\mathcal{S}_i| = s$.

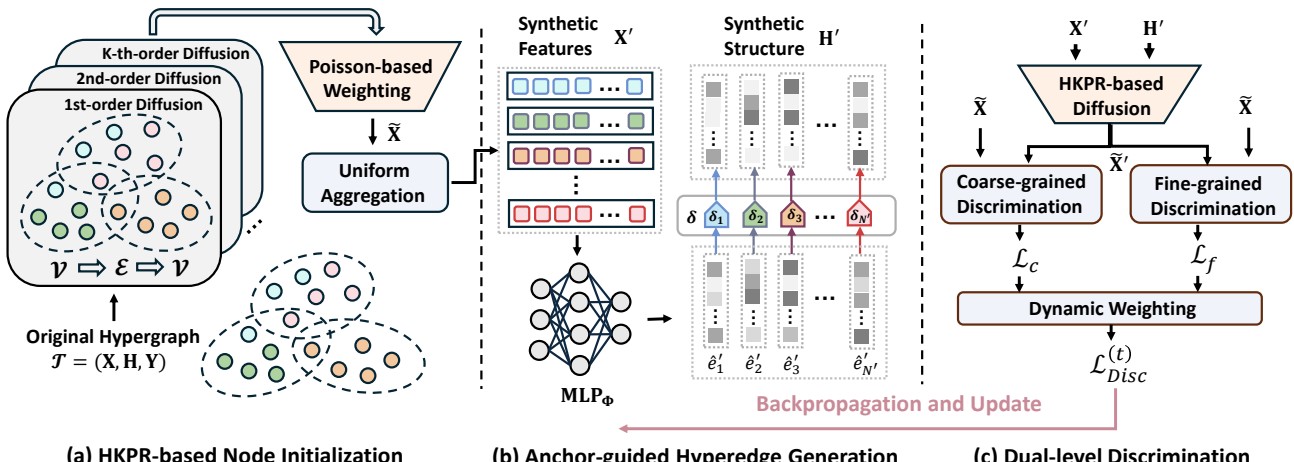

**(a) HKPR-based Node Initialization**  **(b) Anchor-guided Hyperedge Generation**  **(c) Dual-level Discrimination**

*Figure 2.* Overall framework of AHGCDD.

## 3.2. Anchor-guided Hyperedge Generation

As discussed in Section 1, decoupling structure optimization from the utility-preserving refinement process may lead to the synthesis of spurious high-order interactions, thereby degrading data utility. To address this issue, we propose an anchor-guided hyperedge generation strategy, in which hyperedges are generated based on feature-level associations learned by a structure generator. This design naturally enables the joint optimization of condensed features and structure in the subsequent discrimination process (Section 3.3).

Specifically, each synthetic node $v_i'$ is taken in turn as an anchor node. For a given anchor, we induce a weighted hyperedge $\hat{e}_i'$ by quantifying feature-driven associations between the anchor node and all remaining nodes, which are treated as candidate nodes. These associations are modeled by a learnable function, and the resulting connectivity strengths form an incidence score vector $\hat{\mathbf{H}}_i'$, defined as:

$$\begin{aligned}
\hat{\mathbf{H}}_i' &= [\hat{\mathbf{h}}_{i,1}', \hat{\mathbf{h}}_{i,2}', \ldots, \hat{\mathbf{h}}_{i,N'}']^\top, \\
\hat{\mathbf{h}}_{i,j}' &= \mathrm{sigmoid}\big(\mathbf{MLP}_{\boldsymbol{\Phi}}([\mathbf{X}_i'; \mathbf{X}_j'])\big),
\end{aligned} \quad (7)$$

where $\mathbf{MLP}_{\boldsymbol{\Phi}}$ denotes a multi-layer perceptron parameterized by $\boldsymbol{\Phi}$, and $[\cdot; \cdot]$ represents the concatenation operator. To prune weak node–hyperedge connections that can introduce redundant structure and distort message passing in small-scale condensed hypergraphs, we further introduce the anchor-adaptive thresholds $\boldsymbol{\delta}$ as:

$$\boldsymbol{\delta} = [\boldsymbol{\delta}_1, \boldsymbol{\delta}_2, \ldots, \boldsymbol{\delta}_{N'}] \in \mathbb{R}^{N'}. \quad (8)$$

where each $\boldsymbol{\delta}_i$ denotes a learnable threshold for the corresponding hyperedge $\hat{e}_i'$. Unlike existing works that rely on a predefined global threshold (Jin et al., 2022; Gong et al., 2025), our anchor-adaptive thresholds eliminate the need for manual sparsity tuning and enable hyperedge-specific

density adaptation, allowing for more flexible and expressive structure generation. Accordingly, for each hyperedge $\hat{e}_i'$, its sparsified incidence vector $\mathbf{H}_i'$ is computed as:

$$\mathbf{H}_i' = \mathrm{ReLU}\big(\hat{\mathbf{H}}_i' - \boldsymbol{\delta}_i\big). \quad (9)$$

In this manner, we bridge condensed features and high-order structural synthesis, enabling their joint optimization.

## 3.3. Dual-level Discrimination Loss

To preserve the downstream utility of the original data, we propose to refine the synthetic hypergraph using a dual-level discrimination loss that captures category-level information at both coarse and fine granularities. This novel objective eliminates repetitive HNN training during condensation, leading to a significantly more efficient yet task-aware optimization process. The pseudo code of AHGCDD is presented in Algorithm 1 in Appendix A.

Given the condensed features $\mathbf{X}'$ and the anchor-guided synthetic structure $\mathbf{H}'$, we first compute the HKPR-based node representations $\tilde{\mathbf{X}}'$ for $\mathcal{S}$ as:

$$\tilde{\mathbf{X}}' \approx \sum_{k=0}^{K} \frac{e^{-\lambda}\lambda^k}{k!} \cdot \mathbf{P}'^{(k)}\mathbf{X}', \quad (10)$$

where $\mathbf{P}' = \mathbf{D}_v'^{-\frac{1}{2}}\mathbf{H}'\mathbf{D}_e'^{-1}\mathbf{H}'^\top\mathbf{D}_v'^{-\frac{1}{2}}$ denotes the normalized propagation matrix over $\mathcal{S}$, and $\mathbf{D}_v', \mathbf{D}_e'$ are the vertex and hyperedge degree matrices, respectively. To preserve the class-wise distribution of the original data while enabling a global partition of decision boundaries across categories, we design a coarse-grained discrimination loss:

$$\mathcal{L}_c = \sum_{i=1}^{C} \left(1 - \frac{\mathbf{C}_i^\top \cdot \mathbf{C}_i'}{\|\mathbf{C}_i\|\,\|\mathbf{C}_i'\|}\right) + \sum_{i \neq j}^{C} \frac{\mathbf{C}_i^\top \cdot \mathbf{C}_j'}{\|\mathbf{C}_i\|\,\|\mathbf{C}_j'\|}, \quad (11)$$

$$\mathbf{C} = \mathbf{Y}^\top\tilde{\mathbf{X}}, \quad \mathbf{C}' = \mathbf{Y}'^\top\tilde{\mathbf{X}}'$$

where $\mathbf{C}$ and $\mathbf{C}'$ denote the class-level prototypes of $\mathcal{T}$ and $\mathcal{S}$, respectively. While $\mathcal{L}_c$ enforces intra-class alignment and inter-class separability in a global view, it does not explicitly align the instance-level geometry between the original and condensed data within each class, which is crucial for shaping local decision boundaries. To complement $\mathcal{L}_c$, we introduce a fine-grained discrimination loss defined as:

$$
\begin{aligned}
\mathcal{L}_f = - \sum_{i \in \mathcal{V}'} \mathop{\mathbb{E}}_{\substack{p \sim \mathcal{U}(P(i)) \\ Q(i) \sim \mathcal{U}(N(i))}} & \left[ \log \exp\left( \tilde{\mathbf{X}}_i'^{\top} \tilde{\mathbf{X}}_p \right) \right. \\
& \left. - \log \left( \exp\left( \tilde{\mathbf{X}}_i'^{\top} \tilde{\mathbf{X}}_p \right) + \sum_{q \in Q(i)} \exp\left( \tilde{\mathbf{X}}_i'^{\top} \tilde{\mathbf{X}}_q \right) \right) \right],
\end{aligned}
$$

$$
\text{with} \quad P(i) = \{ j \mid y_j = y_i' \}, \; N(i) = \{ j \mid y_j \neq y_i' \} \tag{12}
$$

where $Q(i) \subset N(i)$ with $|Q(i)| = N_{\text{neg}}$ denoting the number of negative samples, and $\mathcal{U}(\cdot)$ denotes uniform sampling. We now provide a detailed explanation of $\mathcal{L}_f$. For each synthetic node $v_i'$, we randomly sample one original node from the same class as a positive example and a set of original nodes from different classes as negative examples. The fine-grained loss encourages the representation of each condensed node to be closer to its positive sample while being separated from negative ones, thereby capturing instance-level discriminative patterns within each class and refining local decision boundaries in the condensed hypergraph.

During optimization, we adopt a coarse-to-fine training strategy, where condensation progresses from *global distribution alignment* to *local structural refinement*. In the early stages, emphasizing the coarse-grained discrimination loss $\mathcal{L}_c$ encourages the condensed data to preserve the class-wise distribution of the original hypergraph, thereby establishing well-separated global decision regions. As training proceeds, the optimization gradually shifts toward the fine-grained loss $\mathcal{L}_f$, allowing condensed nodes to capture instance-level geometry within each class, which is crucial for refining local decision boundaries. Based on this intuition, we formulate a dynamically weighted objective as:

$$
\mathcal{L}_{Disc}^{(t)} = \cos\left( \frac{\pi t}{2T} \right) \mathcal{L}_c + \sin\left( \frac{\pi t}{2T} \right) \mathcal{L}_f, \tag{13}
$$

where $T$ denotes the total number of condensation epochs. As $t$ progresses, $\cos\left( \frac{\pi t}{2T} \right)$ decreases from 1 to 0, while $\sin\left( \frac{\pi t}{2T} \right)$ increases from 0 to 1. This gradual reweighting approach eliminates the need for additional hyperparameters and prevents either objective from dominating the optimization. Overall, this coarse-to-fine optimization strategy stabilizes training and yields a compact yet discriminative condensed dataset that preserves both global class-level separability and local inter-class variation.

**Theoretical Understanding.** We theoretically analyze how the two components of the dual-level discrimination objective preserve the training utility of $\mathcal{S}$.

*(i) Analysis on Coarse-grained Loss $\mathcal{L}_c$.* For the intra-class prototype alignment term in $\mathcal{L}_c$, the following theorem shows that this objective can be interpreted as minimizing the maximum mean discrepancy (MMD) between $\mathcal{S}$ and $\mathcal{T}$ over the joint space of normalized structure-enriched features and labels, thereby explaining its role in encouraging class-wise distribution alignment.

**Theorem 3.3.** *Let $P$ and $Q$ denote the joint distributions of $(\bar{\mathbf{x}}, y)$ induced by the original and condensed data, respectively, where $\bar{\mathbf{x}} = \tilde{\mathbf{x}}/\|\tilde{\mathbf{x}}\|$ denotes the $\ell_2$-normalized features. Define the feature map $\phi(\bar{\mathbf{x}}, y) = \mathbf{e}_y \otimes \bar{\mathbf{x}} \in \mathbb{R}^{Cd}$, where $\mathbf{e}_y$ is the one-hot vector of label $y$ and $\otimes$ denotes the Kronecker product. Consider the linear kernel*

$$
k\big( (\bar{\mathbf{x}}, y), (\bar{\mathbf{x}}', y') \big) = \langle \phi(\bar{\mathbf{x}}, y), \phi(\bar{\mathbf{x}}', y') \rangle, \tag{14}
$$

*minimizing the intra-class alignment term in $\mathcal{L}_c$ leads to the minimization of the squared MMD between $P$ and $Q$:*

$$
\text{MMD}_k^2(P, Q) = \|\mathbb{E}_P[\phi] - \mathbb{E}_Q[\phi]\|_2^2. \tag{15}
$$

The proof of Theorem 3.3 can be found in Appendix E.3. By penalizing similarities between mismatched class prototypes, the second term in $\mathcal{L}_c$ discourages inter-class alignment, thereby enhancing class-level discriminability. To formally characterize this effect, we define a margin-based separability metric, referred to as the *class-level margin*, which quantifies how much more similar a class prototype is to its matched counterpart than to any mismatched one.

**Definition 3.4** (Class-level margin). Let $u_i = \mathbf{C}_i/\|\mathbf{C}_i\|$ and $u_i' = \mathbf{C}_i'/\|\mathbf{C}_i'\|$ be the normalized class prototypes of $\mathcal{T}$ and $\mathcal{S}$ for class $i$. The class-level margin is defined as:

$$
m_i = u_i^{\top} u_i' - \max_{j \neq i} u_i^{\top} u_j', \tag{16}
$$

**Proposition 3.5.** *Assume the total positive cross-class similarity is bounded by $\sum_{i \neq j} [u_i^{\top} u_j']_+ \leq \varepsilon$, where $[x]_+ = \max(x, 0)$. Then the average class-level margin satisfies*

$$
\frac{1}{C} \sum_{i=1}^{C} m_i \;\geq\; \frac{1}{C} \sum_{i=1}^{C} u_i^{\top} u_i' \;-\; \frac{\varepsilon}{C}. \tag{17}
$$

The proof of this proposition is deferred to Appendix E.4.

*Remark* 3.6. Proposition 3.5 indicates that the combined effect of maximizing intra-class similarity and minimizing inter-class similarity enlarges the average inter-class separation, leading to a more globally discriminative representation space. This global separability makes different class patterns easier to distinguish and facilitates downstream training on the condensed data.

*(ii) Analysis on Fine-grained Loss $\mathcal{L}_f$.* For each synthetic node $i$, given a sampled positive original node $p \in P(i)$ and a set of negative original nodes $Q(i) \subset N(i)$, the corresponding fine-grained loss term $l_i$ can be written as:

$$l_i = -\log \frac{\exp(s_{i,p})}{\exp(s_{i,p}) + \sum_{q \in Q(i)} \exp(s_{i,q})}, \; s_{i,j} = \tilde{\mathbf{X}}_i'^{\top} \tilde{\mathbf{X}}_j. \quad (18)$$

Here, $s_{i,j}$ denotes the inner-product similarity between a condensed node $v_i'$ and an original node $v_j$. To characterize the instance-level discriminative behavior of condensed data, we introduce the *Mis-ranking Event*, which determines whether positive samples attain higher similarity scores than negative ones for each synthetic node.

**Definition 3.7** (Mis-ranking Event). For a condensed node $i$ with a positive sample $p$ and a set of negative samples $Q(i)$, the mis-ranking event is defined as:

$$\mathcal{E}_i \triangleq \{ \exists\, q \in Q(i) \; : \; s_{i,q} \geq s_{i,p} \}, \quad (19)$$

which indicates that at least one negative sample is ranked no lower than the positive one.

The following proposition elucidates the relationship between the fine-grained discrimination objective and the probability of the Mis-ranking Event:

**Proposition 3.8.** *Given the node-wise fine-grained discrimination loss $\ell_i$, the probability of the Mis-ranking Event $\mathcal{E}_i$ is upper-bounded by:*

$$\Pr(\mathcal{E}_i) \; \leq \; \mathbb{E}\big[ e^{\ell_i} - 1 \big]. \quad (20)$$

The proof of proposition 3.8 is provided in Appendix E.5.
*Remark* 3.9. Proposition 3.8 implies that minimizing $\ell_i$ directly minimizes an upper bound on the probability that a negative sample outranks the positive one. As a result, $\mathcal{L}_f$ refines local decision boundaries around each condensed node and alleviates confusion caused by negative samples with high similarity scores. This instance-level control enables the condensed data to better preserve the local discriminative geometry of the original data, which cannot be captured by prototype-based alignment alone.

### 3.4. Complexity Analysis

For the node initialization module, the time complexity of computing the HKPR-based diffusion in Eq. (4) is $\mathcal{O}(KM\delta_e d)$, where $\delta_e$ denotes the average hyperedge size. The complexity of uniform aggregation is $\mathcal{O}(N'sd)$, and computing $\mathbf{C} = \mathbf{Y}^{\top}\tilde{\mathbf{X}}$ in Eq. (11) costs $\mathcal{O}(Nd)$. During condensation, anchor-guided hyperedge generation incurs a cost of $\mathcal{O}(L_{\boldsymbol{\Phi}} N'^2 d^2)$, where $L_{\boldsymbol{\Phi}}$ is the number of layers in $\mathbf{MLP}_{\boldsymbol{\Phi}}$. The computation of $\tilde{\mathbf{X}}'$ has a complexity of $\mathcal{O}(KM'\delta_e' d)$, with $\delta_e'$ denoting the average size of condensed hyperedges. The time complexities of $\mathcal{L}_c$ and $\mathcal{L}_f$ are

*Table 1.* Statistics of datasets.

| Dataset | #Nodes | #Edges | $\sum_{e \in E}|e|$ | #Features | #Classes | Description |
|---|---|---|---|---|---|---|
| Cora | 2,708 | 1,579 | 7,494 | 1,433 | 7 | co-citation network |
| Pubmed | 19,717 | 7,963 | 54,346 | 500 | 3 | co-citation network |
| DBLP-CA | 41,302 | 22,363 | 140,863 | 1,425 | 6 | co-authorship network |
| Walmart | 88,860 | 69,906 | 549,490 | 100 | 11 | co-purchase network |
| Yelp | 50,758 | 679,302 | 4,574,352 | 1,862 | 9 | co-occurrence network |
| MAG-PM | 153,009 | 192,180 | 1,138,160 | 256 | 22 | co-authorship network |

$\mathcal{O}(N'd + C^2 d)$ and $\mathcal{O}(N'N_{\text{neg}}d)$, respectively. Considering $T$ optimization iterations, the overall time complexity of AHGCDD is $\mathcal{O}\big(KM\delta_e d + T\big(L_{\boldsymbol{\Phi}}N'^2 d^2 + N'N_{\text{neg}}d\big)\big)$.

## 4. Experiments

In this section, we present a comprehensive empirical evaluation to demonstrate the efficacy and efficiency of AHGCDD. Additional experimental results are provided in Appendix D.

### 4.1. Experimental Settings

**Datasets.** We use six widely used hypergraph benchmarks across various scales, including Cora, Pubmed, DBLP-CA (Yadati et al., 2019), Walmart, Yelp (Chien et al., 2022), and MAG-PM (Sinha et al., 2015). For all datasets, we adopt a split of 50%/25%/25% for training, validation, and testing, following previous HNN studies (Chien et al., 2022; Wang et al., 2023b; Tang et al., 2025). Dataset statistics can be found in Table 1.

**Baselines.** We compare our method with representative baselines, which fall into two categories: **(1) Traditional coreset methods:** Random, Herding (Welling, 2009), and K-Center (Sener & Savarese, 2018); **(2) Hypergraph condensation methods:** HG-Cond and HG-Cond-NHL (Gong et al., 2025). Specifically, HG-Cond-NHL fixes the hyperedges initialized by NHL and optimizes only node features, whereas HG-Cond updates the hypergraph structure during condensation with the pre-trained NHL. Moreover, we extend the recent state-of-the-art structure-free graph condensation method GCPA (Li et al., 2025b) to the hypergraph setting by incorporating hypergraph message passing in the precomputation stage. This extended variant is denoted as HGCPA in our evaluation.

The details of the evaluation protocol and hyperparameter settings are provided in Appendix C.1 and C.2, respectively. Our code is available at https://github.com/Coco-Hut/AHGCDD.

### 4.2. Overall Performance

In this section, we conduct a comprehensive comparison of our AHGCDD against baselines across various reduction ratios for node classification, as presented in Table 2. AHGCDD-X denotes the graph-less variant of AHGCDD, which fixes the condensed structure as the identity matrix $\mathbf{I}$

*Table 2.* Effectiveness evaluation of the condensed data, mean accuracy (%) ± standard deviation. **Bold** indicates the best performance, and underline means the runner-up. OOM denotes the out-of-memory issue.

| Dataset | Ratio ($r$) | Traditional Coreset Methods | | | Hypergraph Condensation Methods | | | | | Whole Dataset |
|---|---|---|---|---|---|---|---|---|---|---|
| | | Random | Herding | K-Center | HGCPA | HG-Cond-NHL | HG-Cond | AHGCDD-X | AHGCDD | |
| Cora | 0.5% | 39.68±2.06 | 41.86±4.57 | 43.07±3.34 | 62.07±3.81 | 71.74±3.72 | 71.43±3.22 | 69.25±1.08 | **74.83±1.95** | 77.90±1.17 |
| | 1% | 43.99±2.76 | 49.45±3.32 | 44.49±2.78 | 63.46±2.90 | 73.91±1.18 | 73.36±3.26 | 72.32±2.68 | **76.48±1.58** | |
| | 2.5% | 53.38±3.92 | 55.33±3.45 | 49.19±2.78 | 64.55±2.05 | 76.72±1.76 | 74.55±2.36 | 75.13±1.52 | **77.85±2.21** | |
| Pubmed | 0.5% | 71.33±1.97 | 75.80±1.16 | 56.48±1.31 | 66.43±4.07 | 76.75±0.36 | 76.92±0.55 | 77.42±0.51 | **78.66±1.57** | 86.17±0.52 |
| | 1% | 75.13±0.92 | 78.01±0.62 | 61.34±1.55 | 68.85±5.62 | 77.15±0.40 | 77.65±0.31 | 78.70±0.63 | **79.57±0.26** | |
| | 2.5% | 77.69±0.82 | 80.42±0.45 | 65.56±0.93 | 69.67±4.38 | 77.83±0.41 | 77.59±1.31 | 78.85±0.45 | **81.09±0.35** | |
| DBLP-CA | 0.1% | 60.38±2.05 | 63.87±3.01 | 54.20±3.94 | 83.63±1.19 | 85.57±0.46 | 85.78±3.14 | 86.34±0.54 | **86.37±0.39** | 90.75±0.26 |
| | 0.5% | 76.42±1.16 | 80.17±0.87 | 69.67±1.65 | 84.72±0.42 | 87.68±0.42 | 86.66±1.89 | 88.14±0.34 | **88.80±0.65** | |
| | 1% | 82.34±1.05 | 84.69±0.47 | 76.27±1.50 | 85.06±0.32 | 88.08±0.27 | 88.30±2.08 | 88.36±0.28 | **89.06±0.57** | |
| Walmart | 0.1% | 45.93±2.35 | 47.40±1.16 | 45.03±2.31 | 47.55±1.01 | 50.27±1.52 | 50.31±4.38 | 57.66±0.65 | **62.77±0.60** | 75.22±0.39 |
| | 0.5% | 50.65±1.36 | 55.96±1.33 | 47.88±1.83 | 48.61±0.70 | 56.73±1.09 | 54.13±5.44 | 63.67±0.53 | **67.92±0.66** | |
| | 1% | 54.34±2.01 | 60.66±1.85 | 51.28±3.03 | 50.57±0.88 | 59.65±1.22 | 56.68±3.75 | 66.06±0.71 | **69.09±0.34** | |
| Yelp | 0.05% | 21.82±1.56 | 21.78±3.02 | 18.84±1.94 | 22.60±1.27 | OOM | OOM | 23.67±1.23 | **25.43±0.54** | 33.71±0.24 |
| | 0.25% | 22.79±1.71 | 23.01±1.76 | 21.78±3.75 | 23.75±4.71 | OOM | OOM | 24.64±1.22 | **26.80±0.31** | |
| | 0.5% | 25.27±0.37 | 23.83±1.65 | 22.53±0.96 | 25.93±0.88 | OOM | OOM | 26.19±0.23 | **27.09±0.06** | |
| MAG-PM | 0.05% | 26.38±3.76 | 30.29±1.72 | 23.20±4.16 | 36.22±4.12 | 53.13±0.29 | 50.83±2.76 | **53.35±0.46** | 53.16±0.89 | 62.58±0.05 |
| | 0.25% | 40.50±0.96 | 43.79±0.80 | 35.87±0.97 | 44.82±3.83 | 55.37±0.35 | 53.07±1.82 | 56.60±0.26 | **57.55±0.54** | |
| | 0.5% | 43.38±1.03 | 47.92±0.30 | 41.11±1.18 | 47.91±3.54 | 56.25±0.29 | 52.10±2.33 | 57.17±0.37 | **58.89±0.61** | |

and refines only the synthetic features. We have the following key observations: (1) Overall, AHGCDD achieves superior performance compared to the baselines. The condensed data is highly comparable to the original dataset when used to train HNNs for node classification. (2) Coreset selection methods generally underperform condensation-based approaches across most settings. In addition, HGCPA performs markedly worse than carefully designed HGC methods in most cases, indicating that existing state-of-the-art structure-free GC methods do not generalize well to hypergraph learning scenarios. These findings highlight the necessity of developing condensation methods specifically tailored for HNNs. (3) HG-Cond-NHL achieves performance comparable to, or even better than, HG-Cond under different experimental settings, and HG-Cond exhibits substantially larger performance variance across runs. This is because the pre-trained structure generator NHL is not jointly optimized with the condensed node features to preserve training utility. As a result, the structure updates during condensation may become misaligned with the evolving features, introducing spurious high-order interactions and ultimately degrading data utility. (4) AHGCDD outperforms its structure-free variant in nearly all settings, highlighting the importance of preserving hypergraph structure in condensed data for effective HNN training.

### 4.3. Efficiency Comparison

We now compare the running time and memory overhead of AHGCDD and HG-Cond during condensation. As shown in Table 3, our AHGCDD is significantly more efficient than HG-Cond, achieving speedups ranging from 28× to 144× across different datasets. For instance, condensing

*Table 3.* Efficiency comparison between HG-Cond and AHGCDD in terms of condensation time and memory consumption.

| Dataset | HG-Cond | | AHGCDD | |
|---|---|---|---|---|
| | Time (s) | Mem (MB) | Time (s) | Mem (MB) |
| Cora ($r$=1%) | 519.2 | 1095 | 3.6 | 362.9 |
| Pubmed ($r$=1%) | 599.4 | 1553 | 11.5 | 1063 |
| DBLP-CA ($r$=0.5%) | 617.5 | 3651 | 13.9 | 3101 |
| Walmart ($r$=0.5%) | 827.9 | 10431 | 29.2 | 1529 |
| Yelp ($r$=0.25%) | OOM | OOM | 132.2 | 8379 |
| MAG-PM ($r$=0.25%) | 1111.6 | 13137 | 27.8 | 2905 |

the large-scale MAG-PM dataset requires over 1100 seconds with HG-Cond, whereas AHGCDD completes the process within 30 seconds. This is mainly because, unlike HG-Cond, our framework eliminates the need for parameter matching and gradient matching as well as repeated training trajectory computations during condensation. Instead, AHGCDD adopts a lightweight discrimination loss, which significantly reduces computational overhead. Moreover, our framework is substantially more memory-efficient: HG-Cond incurs over 10GB of GPU memory consumption on both Walmart and MAG-PM, while AHGCDD requires less than 3GB. On the large-scale and hyperedge-dense Yelp dataset, HG-Cond suffers from out-of-memory (OOM) issues, whereas AHGCDD runs successfully, demonstrating its superior scalability. The reduced memory consumption mainly stems from the simplified architecture of AHGCDD, which avoids training a variational-inference-based neural hyperedge linker with substantial memory overhead.

### 4.4. Ablation Study

**Ablation on Node Initialization.** We first investigate the initialization phase under different information propagation

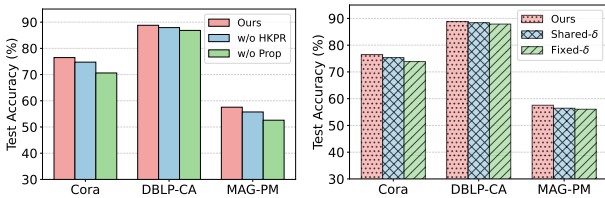

*(a)* Node initialization analysis    *(b)* Sparsity control analysis

*Figure 3.* Effect of HKPR-based node initialization and anchor-adaptive sparsity threshold, where $r$ is set to 1%, 0.5%, and 0.25% for Cora, DBLP-CA, and MAG-PM, respectively.

*Table 4.* Ablation study on discrimination loss.

| Dataset | w/o $\mathcal{L}_c$ | w/o $\mathcal{L}_f$ | w/o Dynamic | **Ours** |
|---|---|---|---|---|
| Cora ($r$=1%) | 72.67±1.33 | 74.15±1.18 | 75.63±1.42 | **76.48±1.58** |
| DBLP-CA ($r$=0.5%) | 87.40±0.98 | 86.93±0.32 | 87.79±0.55 | **88.80±0.65** |
| MAG-PM ($r$=0.25%) | 56.19±0.43 | 55.97±0.25 | 56.84±0.81 | **57.55±0.54** |

strategies. Specifically, *w/o HKPR* replaces HKPR with vanilla hypergraph message passing that does not use Poisson weights, whereas *w/o Prop* completely removes structural information propagation. In Figure 3(a), we observe that our HKPR-based method consistently outperforms the variant with vanilla hypergraph message passing, demonstrating that the proposed path-adaptive diffusion mechanism effectively denoises propagated information and leads to performance improvements. Meanwhile, removing information propagation results in a more pronounced performance degradation than *w/o HKPR*, highlighting the crucial role of structure-aware node initialization in capturing comprehensive node semantics and improving data utility.

**Ablation on Anchor-adaptive Threshold.** To study the contribution of our anchor-adaptive threshold, we compare it with two variants: *Shared-δ*, which employs a single learnable threshold for all anchors, and *Fixed-δ*, which uses a predefined constant threshold selected from $\{0.1, 0.2, \dots, 0.9\}$. As shown in Figure 3(b), the *Fixed-δ* variant shows noticeably lower performance than the two learnable variants across all datasets. This is because learnable thresholds can incorporate discriminative knowledge during optimization and expand the search space. Moreover, the proposed anchor-adaptive threshold achieves superior performance over the global learnable threshold (i.e., *Shared-δ*), demonstrating that allowing anchor-specific hyperedge densities enables more flexible and fine-grained sparsity control.

**Ablation on Discrimination Loss.** Table 4 evaluates the impact of $\mathcal{L}_c$, $\mathcal{L}_f$, and the dynamic weighting strategy. We observe that removing any individual component results in a noticeable performance degradation, underscoring the necessity of each technique in the condensation procedure. More concretely, using either the coarse-grained loss or the fine-grained loss alone leads to inferior performance, indicating that these two objectives provide complementary

discriminative signals. In addition, the dynamic weighting strategy consistently outperforms static weighting, further demonstrating that our coarse-to-fine optimization trajectory facilitates more effective convergence of the condensed data.

## 5. Related Work

In this section, we review related work on hypergraph size reduction, while the discussion of graph condensation is deferred to Appendix B.2.

**Hypergraph Size Reduction.** Existing hypergraph reduction methods can be broadly categorized into three classes: hypergraph coreset, coarsening, and condensation. Coreset methods (Welling, 2009; Sener & Savarese, 2018) select a subset of original nodes while retaining the induced hyperedges. Coarsening approaches (Aghdaei et al., 2021; Aghdaei & Feng, 2022) reduce hypergraph size by clustering nodes into supernodes, aiming to preserve global structural properties for more efficient downstream processing. For instance, HyperSF (Aghdaei et al., 2021) adopts a local max-flow-based clustering strategy to minimize ratio cuts, whereas HyperEF (Aghdaei & Feng, 2022) decomposes large hypergraphs into clusters with minimal inter-cluster hyperedges. However, coarsening-based methods are generally limited to structure-only hypergraphs and are not applicable to attributed hypergraphs, preventing them from leveraging feature semantics. HG-Cond (Gong et al., 2025) makes the first attempt to explore hypergraph condensation. The framework first trains a Neural Hyperedge Linker to capture high-order connectivity patterns via variational inference. Then, with the optimized structure generator, it utilizes a Gradient-Parameter Synergistic Matching objective to refine synthetic hypergraphs. Although this design significantly outperforms coreset- and coarsening-based methods, the decoupled architecture may result in the generation of misleading high-order interactions, and the gradient–parameter matching incurs substantial computational overhead.

## 6. Conclusion

In this paper, we present AHGCDD, a novel framework for efficient and effective hypergraph condensation. It addresses the optimization misalignment issue in prior work by introducing an anchor-guided hyperedge generation scheme. This strategy synthesizes density-adaptive hyperedges via HKPR-based higher-order node features, enabling unified optimization of condensed features and structure. Moreover, we propose a dual-level discrimination loss with theoretical justification, which aligns the training utility of the condensed and original hypergraphs while eliminating the need for repetitive training trajectory optimization. Extensive experiments on six real-world benchmarks verify the superiority of AHGCDD in terms of effectiveness and efficiency.

## Acknowledgements

Xiaoyang Wang is supported by ARC DP240101322 and DP260100689. Wenjie Zhang is supported by the Australian Research Council Centre of Excellence for Mathematical Modelling of Cellular Systems CE230100001.

## Impact Statement

This paper presents work whose goal is to advance the field of machine learning. There are many potential societal consequences of our work, none of which we feel must be specifically highlighted here.

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

# A. Algorithm

We show the detailed algorithm of AHGCDD in Algorithm 1. Specifically, we first set the condensed label set $\mathbf{Y}'$ to fixed values and initialize $\mathbf{X}'$ with the HKPR-based initialization method (lines 3-5). In each condensation loop, we first synthesize the sparsified hypergraph structure $\mathbf{H}'$ via the anchor-guided hyperedge generation strategy (line 7), and then utilize HKPR-based diffusion to compute the structure-aware node features $\tilde{\mathbf{X}}'$ for the condensed data (line 8). Then, we calculate the weighted sum of coarse-grained and fine-grained discrimination loss (lines 9-11). The losses are backpropagated to alternately update the condensed features $\mathbf{X}'$ and the structure generator, where the latter includes $\mathbf{MLP_\Phi}$ and the anchor-adaptive threshold $\boldsymbol{\delta}$ (lines 12-17). When finishing the updating of condensed hypergraph parameters, we apply anchor-guided hyperedge generation to construct the final structure (line 19) and return the condensed data (line 20).

---

**Algorithm 1** AHGCDD for Hypergraph Data Condensation

---

1: **Input:** Original hypergraph $\mathcal{T} = (\mathbf{X}, \mathbf{H}, \mathbf{Y})$
2: **Require:** Training epochs $T$, learning rate $\eta_1, \eta_2$, number of update iterations $\tau_1, \tau_2$.
3: Initialize labels $\mathbf{Y}'$ with the same distribution as $\mathbf{Y}$
4: Compute HKPR-based features $\tilde{\mathbf{X}}$ for $\mathcal{T}$ via Eq. (4)
5: Initialize condensed features $\mathbf{X}'$ via Eq. (6)
6: **for** $t = 0$ to $T - 1$ **do**
7:     Generate structure $\mathbf{H}'$ via Eq. (7) - (9)
8:     Compute HKPR-based features $\tilde{\mathbf{X}}'$ for $\mathcal{S}$ via Eq. (10)
9:     Compute coarse-grained loss $\mathcal{L}_c$ via Eq. (11)
10:     Compute fine-grained loss $\mathcal{L}_f$ via Eq. (12)
11:     Compute weighted discrimination loss $\mathcal{L}_{Disc}^{(t)} = \cos\left(\frac{\pi t}{2T}\right)\mathcal{L}_c + \sin\left(\frac{\pi t}{2T}\right)\mathcal{L}_f$
12:     **if** $t\%(\tau_1 + \tau_2) < \tau_1$ **then**
13:         Update $\mathbf{X}' \leftarrow \mathbf{X}' - \eta_1 \nabla_{\mathbf{X}'} \mathcal{L}_{Disc}^{(t)}$
14:     **else**
15:         Update $\mathbf{\Phi} \leftarrow \mathbf{\Phi} - \eta_2 \nabla_{\mathbf{\Phi}} \mathcal{L}_{Disc}^{(t)}$
16:         Update $\boldsymbol{\delta} \leftarrow \boldsymbol{\delta} - \eta_2 \nabla_{\boldsymbol{\delta}} \mathcal{L}_{Disc}^{(t)}$
17:     **end if**
18: **end for**
19: Generate structure $\mathbf{H}'$ via Eq. (7) - (9)
20: **Output:** Condensed hypergraph $\mathcal{S} = (\mathbf{X}', \mathbf{H}', \mathbf{Y}')$

---

# B. Background Knowledge

## B.1. Hypergraph Neural Networks

Hypergraphs generalize pairwise graphs by allowing each hyperedge to connect an arbitrary number of nodes, enabling the modeling of higher-order relations (Tan et al., 2023; Luo et al., 2024). HNNs have emerged as a powerful tool for representation learning on hypergraphs (Yadati et al., 2019; Prokopchik et al., 2022; Wang et al., 2023b; Saxena et al., 2024; Tang et al., 2025; Li et al., 2026a). Typically, HNNs adopt a two-stage message-passing scheme in each layer, where hyperedge representations are first updated by aggregating information from their incident nodes, followed by updating node representations through aggregation from the connected hyperedges:

$$
\begin{aligned}
z_{e,j}^{(l)} &= f_{\mathcal{V} \to \mathcal{E}}^{(l)}\left(z_{e,j}^{(l-1)}, \{z_{v,k}^{(l-1)} | v_k \in e_j\}\right) \\
z_{v,i}^{(l)} &= f_{\mathcal{E} \to \mathcal{V}}^{(l)}\left(z_{v,i}^{(l-1)}, \{z_{e,k}^{(l)} | v_i \in e_k\}\right).
\end{aligned}
\tag{21}
$$

Here, $z_{e,j}^{(l)}$ and $z_{v,i}^{(l)}$ are the embeddings of $e_j$ and $v_i$ at layer $l$. The functions $f_{\mathcal{V} \to \mathcal{E}}^{(l)}$ and $f_{\mathcal{E} \to \mathcal{V}}^{(l)}$ aggregate information from nodes and hyperedges, respectively. Due to their strong ability to capture higher-order information, HNNs have demonstrated state-of-the-art performance across a wide range of industrial and scientific applications, such as product recommendation (Khan et al., 2025), 3D object detection (Fixelle, 2025), and disease diagnosis (Han et al., 2025).

## B.2. Related Works on Graph Condensation

Graphs model pairwise relations and serve as a fundamental structure for relational learning. (Zhai et al., 2025b;a; Tan et al., 2026a;b). Graph condensation (GC) aims to synthesize a compact graph that preserves the training utility of the original graph. Existing GC methods can be broadly categorized into structure-based and structure-free approaches. Structure-based methods explicitly model graph structure during condensation. GCond (Jin et al., 2022) parameterizes the graph structure as a function of learnable node features and optimizes it via gradient matching. SGDD (Yang et al., 2023a) enhances GCond by incorporating original structural information, thereby improving generalization. GEDM (Liu et al., 2024a) mitigates spectral bias through an eigenbasis matching objective, enabling better cross-architecture generalization. EXGC (Fang et al., 2024) improves efficiency by adopting a mean-field variational approximation and pruning redundant parameters, while DisCo (Xiao et al., 2025) reformulates GC into a two-stage, GNN-free pipeline that independently condenses nodes and generates edges. CGC (Gao et al., 2025a) further proposes a training-free framework that transforms class-level distribution matching into a class-partition problem, enabling efficient EM-based clustering. Structure-free methods bypass explicit structure generation and directly synthesize compact node features. SFGC (Zheng et al., 2023) and GEOM (Zhang et al., 2024) perform condensation by matching the training trajectories of the original graph, whereas GCPA (Li et al., 2025b) introduces a precomputation stage with diversity-aware adaptation to simplify training and achieve SOTA performance.

# C. More Experimental Details

## C.1. Evaluation Protocol

To fairly evaluate the quality of condensed data, we follow a two-step protocol (Xu et al., 2026) for all methods: (1) **Condensation step**, where different distillation methods are applied to the original training hypergraphs to generate synthetic data; and (2) **Evaluation step**, where HNNs are trained from scratch on the synthetic hypergraphs and evaluated on the real test sets. The representative HGNN (Feng et al., 2019) is used as the evaluation model. We condense each dataset to three different condensation ratios ($r$), defined as the ratio of the number of synthetic nodes $rN$ ($0 < r < 1$) to the number of original nodes $N$. Specifically, we set $r$ to $\{0.5\%, 1\%, 2.5\%\}$ for small-scale datasets (Cora and Pubmed), $\{0.1\%, 0.5\%, 1\%\}$ for medium-scale datasets (DBLP-CA and Walmart), and $\{0.05\%, 0.25\%, 0.5\%\}$ for large-scale datasets (Yelp and MAG-PM). For each setting, we generate five synthetic hypergraphs, evaluate each of them five times, and report the average node classification accuracy along with the standard deviation. All experiments are conducted on a Linux server equipped with an Intel Xeon Silver 4208 CPU, a Quadro RTX 6000 GPU with 24GB of memory, and 128GB of RAM.

## C.2. Hyperparameter Settings

The structure function $\mathbf{MLP_{\Phi}}$ is a 3-layer MLP with 256 hidden units. The path weight $\lambda$ is selected from $\{1, 2, 3, 4, 5\}$. The number of negative samples $N_{\text{neg}}$ is searched over $\{1, 5, 10, 20, 50\}$. The subset size $s$ is chosen from $\{5, 10, 20\}$, and the number of synthetic training epochs $T$ is tuned from $\{50, 100, 150, 200\}$. The learning rates for updating the condensed features and structure, denoted by $\eta_1$ and $\eta_2$, are selected from $\{0.01, 0.001, 0.0001\}$. The update intervals for feature and structure optimization are set to $\tau_1 = 5$ and $\tau_2 = 15$, respectively. We use the DHG-Bench (Li et al., 2026b) library to implement different HNN models for evaluation. All learnable parameters are optimized using the Adam optimizer, and all hyperparameters are tuned based on the validation set.

# D. More Experimental Results

## D.1. Cross-architecture Transferability

We evaluate the cross-architecture transferability of different methods by testing the synthetic hypergraphs on five widely used HNN models, including HGNN (Feng et al., 2019), HCHA (Bai et al., 2021), UniGCNII (Huang & Yang, 2021), AllSet (Chien et al., 2022), and ED-HNN (Wang et al., 2023a). As shown in Table 5, AHGCDD achieves the highest average accuracy across all datasets, indicating strong transferability and robustness. Moreover, it significantly reduces the performance gap among different HNNs, as evidenced by the lowest performance variance on all datasets except MAG-PM. This strong generalization can be attributed to two main factors: (1) our framework optimizes condensed data without relying on specific HNN architectures. (2) the condensed features and structures are jointly optimized to align with downstream task semantics, thereby avoiding spurious high-order interactions that could otherwise be amplified by different hypergraph message-passing mechanisms. In addition, we observe that HGCPA exhibits the highest accuracy variance across different architectures, indicating that structure-free GC methods do not generalize well to HNN architectures.

*Table 5.* Cross-architecture transferability of condensed data. Avg. and Std. represent the mean test accuracy and the corresponding standard deviation computed across different evaluated HNNs.

| Dataset | Method | HGNN | HCHA | UniGCNII | AllSet | ED-HNN | Avg. | Std. |
|---|---|---|---|---|---|---|---|---|
| Cora (r=1%) | Herding | 49.45 | 45.61 | 46.20 | 45.46 | 47.53 | 46.85 | 1.49 |
| | HGCPA | 63.46 | 61.40 | 70.11 | 68.04 | 61.84 | 64.79 | 3.48 |
| | HG-Cond | 73.36 | 74.89 | 65.73 | 71.05 | 73.71 | 71.75 | 3.26 |
| | AHGCDD | 76.48 | 77.96 | 75.59 | 76.33 | 77.07 | **76.69** | **0.79** |
| Pubmed (r=1%) | Herding | 78.01 | 78.43 | 78.68 | 78.74 | 78.33 | 78.44 | **0.26** |
| | HGCPA | 68.85 | 68.48 | 72.24 | 68.97 | 61.51 | 68.01 | 3.52 |
| | HG-Cond | 77.65 | 78.66 | 77.98 | 75.03 | 71.75 | 76.21 | 2.55 |
| | AHGCDD | 79.57 | 79.40 | 82.67 | 78.57 | 79.08 | **79.86** | 1.45 |
| DBLP-CA (r=0.5%) | Herding | 80.17 | 79.44 | 78.21 | 74.32 | 79.12 | 78.25 | 2.06 |
| | HGCPA | 84.72 | 83.32 | 86.37 | 77.32 | 80.92 | 82.53 | 3.16 |
| | HG-Cond | 86.66 | 86.50 | 85.45 | 80.62 | 85.56 | 84.96 | 2.22 |
| | AHGCDD | 88.80 | 88.51 | 88.60 | 86.77 | 88.27 | **88.18** | **0.73** |
| Walmart (r=0.5%) | Herding | 55.96 | 57.05 | 50.88 | 56.75 | 58.86 | 55.90 | 2.68 |
| | HGCPA | 48.61 | 48.05 | 42.84 | 54.49 | 63.53 | 51.50 | 7.06 |
| | HG-Cond | 54.13 | 55.02 | 48.67 | 51.58 | 49.19 | 51.72 | 2.55 |
| | AHGCDD | 67.92 | 68.28 | 63.76 | 67.97 | 67.82 | **67.15** | **1.70** |
| Yelp (r=0.25%) | Herding | 23.01 | 22.47 | 25.03 | 23.40 | 24.49 | 23.68 | 0.95 |
| | HGCPA | 23.75 | 23.87 | 24.20 | 20.18 | 25.92 | 23.58 | 1.87 |
| | HG-Cond | OOM | OOM | OOM | OOM | OOM | OOM | OOM |
| | AHGCDD | 26.80 | 26.55 | 26.71 | 26.59 | 27.09 | **26.75** | **0.19** |
| MAG-PM (r=0.25%) | Herding | 43.79 | 41.95 | 41.02 | 40.05 | 37.66 | 40.89 | **2.03** |
| | HGCPA | 44.82 | 45.79 | 35.13 | 35.34 | 27.58 | 37.73 | 6.79 |
| | HG-Cond | 53.07 | 51.49 | 37.57 | 45.44 | 39.89 | 45.49 | 6.12 |
| | AHGCDD | 57.55 | 55.87 | 46.55 | 48.67 | 45.36 | **50.80** | 4.97 |

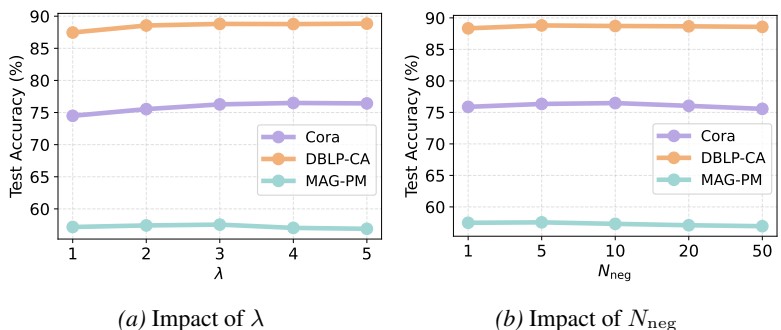

*(a)* Impact of $\lambda$      *(b)* Impact of $N_{\text{neg}}$

*Figure 4.* Sensitivity analysis of hyperparameters.

## D.2. Parameter Sensitivity Analysis

We conduct a parameter analysis to further investigate the influence of $\lambda$ and $N_{\text{neg}}$, as illustrated in Figure 4. Specifically, increasing $\lambda$ initially leads to consistent performance improvements. However, the marginal gains gradually diminish as $\lambda$ increases, and a slight performance degradation can be observed when $\lambda$ exceeds a certain threshold on some datasets. This suggests that while broader information diffusion is beneficial, excessively large diffusion ranges may introduce excessive smoothing, revealing an inherent trade-off in the choice of $\lambda$. On the other hand, performance remains stable when varying the $N_{\text{neg}}$, demonstrating the robustness of our framework. Notably, optimal performance can already be achieved with small values (e.g., 5 or 10), which further validates the efficiency advantage of our discrimination loss.

## D.3. Evaluation on Heterophilic Hypergraphs

We further evaluate on three heterophilic datasets (Actor, Amazon-ratings, and Pokec (Li et al., 2025a)) with a condensation ratio of $r = 1\%$. Table 6 shows that our method consistently achieves state-of-the-art performance and preserves over 90%

of the full-data performance across all datasets, demonstrating strong robustness and generalizability in heterophilic settings.

*Table 6.* Performance comparison on heterophilic datasets.

| Dataset | Herding | HGCPA | HG-Cond | AHGCDD | Whole |
|---|---|---|---|---|---|
| Actor | 70.28±1.06 | 66.40±3.01 | 71.64±0.57 | **72.29±1.21** | 77.83±0.37 |
| Amazon-ratings | 24.16±0.31 | 23.12±1.86 | 24.53±0.77 | **25.41±0.39** | 28.05±0.28 |
| Pokec | 53.03±0.98 | 52.37±0.76 | 53.55±0.21 | **55.04±0.47** | 57.87±0.76 |

## D.4. Additional Ablation Study on Dynamic Weighting

We further investigate the effectiveness of different dynamic weighting strategies in the proposed dual-level discrimination objective. Besides the cosine schedule adopted in the main paper, we additionally evaluate two monotonic coarse-to-fine schedules, namely *Linear* and *Step*. For the Linear schedule, the coarse-level weight gradually decreases during training:

$$w_c(t) = 1 - \frac{t}{T}, \qquad w_f(t) = \frac{t}{T}, \tag{22}$$

where $T$ denotes the total number of training iterations. For the Step schedule, the optimization abruptly switches from coarse-level to fine-level objectives at the midpoint:

$$w_c(t) = \begin{cases} 1, & t < \frac{T}{2}, \\ 0, & t \geq \frac{T}{2}, \end{cases} \qquad w_f(t) = 1 - w_c(t). \tag{23}$$

As shown in Table 7, monotonic reweighting generally outperforms static weighting, supporting the effectiveness of our coarse-to-fine design. Linear and Cosine perform best, while Step is less stable, likely due to the abrupt transition at $t = \frac{T}{2}$, indicating the importance of smooth scheduling. Cosine further improves over Linear, possibly because its slower early-stage decay of $w_c(t)$ better preserves global separability.

*Table 7.* Performance comparison of different reweighting schedules.

| Dataset | Static | Linear | Step | Cosine |
|---|---|---|---|---|
| Cora ($r$=1%) | 75.63±1.42 | 75.92±1.18 | 75.33±1.34 | **76.48±1.58** |
| DBLP-CA ($r$=0.5%) | 87.79±0.55 | 88.36±0.52 | 87.97±0.47 | **88.80±0.65** |
| MAG-PM ($r$=0.25%) | 56.84±0.81 | 57.22±0.80 | 56.43±2.48 | **57.55±0.54** |

## D.5. Additional Downstream Tasks

*Table 8.* Node clustering results measured by NMI (%).

| Dataset | Herding | HG-Cond | AHGCDD | Whole |
|---|---|---|---|---|
| Cora ($r$=1%) | 23.27±3.25 | 43.75±1.86 | **47.58±1.84** | 50.46±2.55 |
| DBLP-CA ($r$=0.5%) | 63.23±0.62 | 70.46±1.70 | **71.99±0.67** | 75.10±0.50 |
| MAG-PM ($r$=0.25%) | 18.53±0.59 | 24.56±0.16 | **26.95±0.32** | 31.48±0.75 |

*Table 9.* Node retrieval results measured by MAP (%).

| Dataset | Herding | HG-Cond | AHGCDD | Whole |
|---|---|---|---|---|
| Cora ($r$=1%) | 38.18±1.36 | 67.13±0.70 | **70.95±0.64** | 79.06±0.65 |
| DBLP-CA ($r$=0.5%) | 73.64±0.52 | 79.66±0.84 | **81.65±0.75** | 83.79±0.21 |
| MAG-PM ($r$=0.25%) | 24.48±0.68 | 28.44±0.68 | **31.42±0.95** | 33.88±0.50 |

We conduct additional experiments on node clustering and node retrieval (Lee et al., 2024), measured by NMI (Normalized Mutual Information) and MAP (Mean Average Precision), respectively. The results in Tables 8 and 9 show that AHGCDD consistently achieves superior performance across different downstream tasks and preserves over 85% of the full-data performance, demonstrating strong generalization capability.

# E. Theoretical Analysis

## E.1. Proof of Theorem 3.1

**Theorem E.1.** *Let $\mathcal{L} = \mathbf{I} - \mathbf{P} \in \mathbb{R}^{n \times n}$ denote the normalized hypergraph Laplacian, with eigenvalues $0 = \mu_1 \leq \mu_2 \leq \cdots \leq \mu_n \leq 2$. The HKPR-based diffusion $\tilde{\mathbf{X}} = \mathbf{P}_\lambda \mathbf{X}$ can be expressed as a spectral filtering operation in the hypergraph Fourier domain:*

$$\tilde{\mathbf{X}} = \mathbf{U} \, g(\mathbf{M}) \, \mathbf{U}^\top \mathbf{X}, \tag{24}$$

*where $\mathbf{U} = [\mathbf{u}_1, \ldots, \mathbf{u}_n]$ is the orthonormal eigenbasis of $\mathcal{L}$, $\mathbf{M} = \mathrm{diag}(\mu_1, \ldots, \mu_n)$ is the Laplacian spectrum, and $g(\mu) = e^{-\lambda\mu}$ is an exponentially decaying low-pass filter.*

*Proof.* The HKPR diffusion matrix $\mathbf{P}_\lambda$ can be rewritten in closed form as

$$\mathbf{P}_\lambda = \sum_{k=0}^\infty \frac{e^{-\lambda}\lambda^k}{k!} \mathbf{P}^k = e^{-\lambda}e^{\lambda\mathbf{P}} = e^{-\lambda(\mathbf{I}-\mathbf{P})} = e^{-\lambda\mathcal{L}}. \tag{25}$$

Let the eigendecomposition of the Laplacian be

$$\mathcal{L} = \mathbf{U} \, \mathrm{diag}(\mu_1, \ldots, \mu_n) \, \mathbf{U}^\top, \quad \mathbf{U}\mathbf{U}^\top = \mathbf{I}. \tag{26}$$

Then we have

$$\begin{aligned}
e^{-\lambda\mathcal{L}} &= \sum_{k=0}^\infty \frac{(-\lambda)^k}{k!} \, \mathcal{L}^k = \sum_{k=0}^\infty \frac{(-\lambda)^k}{k!} \, \mathbf{U}\mathbf{\Lambda}^k\mathbf{U}^\top \\
&= \mathbf{U} \left( \sum_{k=0}^\infty \frac{(-\lambda)^k}{k!} \, \mathbf{\Lambda}^k \right) \mathbf{U}^\top = \mathbf{U} \, \mathrm{diag}(e^{-\lambda\mu_1}, \ldots, e^{-\lambda\mu_n}) \, \mathbf{U}^\top.
\end{aligned} \tag{27}$$

Substituting (27) into $\tilde{\mathbf{X}} = \mathbf{P}_\lambda\mathbf{X}$, we obtain

$$\tilde{\mathbf{X}} = \mathbf{U} \, \mathrm{diag}(e^{-\lambda\mu_1}, \ldots, e^{-\lambda\mu_n}) \, \mathbf{U}^\top \mathbf{X}. \tag{28}$$

Therefore, the smoothing operation admits a spectral filtering form, where the filter function is

$$g_\lambda(\mu) = e^{-\lambda\mu}. \tag{29}$$

This completes the proof. $\square$

## E.2. Proof of Lemma 3.2

**Lemma E.2.** *Let $N \sim \mathrm{Poisson}(\lambda)$ with parameter $\lambda > 0$. Then for any $t > 0$, we have*

$$\mathbb{P}\Big[N \geq \lambda + t\sqrt{\lambda}\Big] \leq \exp\Big(-\frac{t^2}{2 + t/\sqrt{\lambda}}\Big). \tag{30}$$

*Proof.* From (Goldreich, 2017), for any $x > 0$, a Poisson random variable $N \sim \mathrm{Poisson}(\lambda)$ satisfies the following Chernoff bounds:

$$\Pr[N \geq \lambda + x] \leq \exp\Big(-\frac{x^2}{2(\lambda + x)}\Big). \tag{31}$$

By substituting $x = t\sqrt{\lambda}$ into the above inequality, we obtain

$$\Pr\Big[N \geq \lambda + t\sqrt{\lambda}\Big] \leq \exp\Big(-\frac{t^2}{2 + t/\sqrt{\lambda}}\Big), \tag{32}$$

which completes the proof. $\square$

### E.3. Proof of Theorem 3.3

**Theorem E.3.** *Let $P$ and $Q$ denote the joint distributions of $(\bar{\mathbf{x}}, y)$ induced by the original and condensed data, respectively, where $\bar{\mathbf{x}} = \hat{\mathbf{x}}/\|\hat{\mathbf{x}}\|$ denotes the $\ell_2$-normalized features. Define the feature map $\phi(\bar{\mathbf{x}}, y) = \mathbf{e}_y \otimes \bar{\mathbf{x}} \in \mathbb{R}^{Cd}$, where $\mathbf{e}_y$ is the one-hot vector of label $y$ and $\otimes$ denotes the Kronecker product. Consider the linear kernel*

$$k\big((\bar{\mathbf{x}}, y), (\bar{\mathbf{x}}', y')\big) = \langle \phi(\bar{\mathbf{x}}, y), \phi(\bar{\mathbf{x}}', y') \rangle, \tag{33}$$

*minimizing the intra-class alignment term in $\mathcal{L}_c$ leads to the minimization of the squared MMD between $P$ and $Q$:*

$$\mathrm{MMD}_k^2(P, Q) = \|\mathbb{E}_P[\phi] - \mathbb{E}_Q[\phi]\|_2^2. \tag{34}$$

*Proof.* By definition of the maximum mean discrepancy (MMD) with a linear kernel, we have

$$\mathrm{MMD}_k^2(P, Q) = \|\mathbb{E}_P[\phi(\bar{\mathbf{x}}, y)] - \mathbb{E}_Q[\phi(\bar{\mathbf{x}}, y)]\|_2^2. \tag{35}$$

We first compute the expectation under the distribution $P$. By the law of total expectation,

$$\mathbb{E}_P[\phi(\bar{\mathbf{x}}, y)] = \sum_{i=1}^{C} \Pr_P(y = i)\, \mathbb{E}_P[\mathbf{e}_i \otimes \bar{\mathbf{x}} \mid y = i] = \sum_{i=1}^{C} \pi_i\, (\mathbf{e}_i \otimes \boldsymbol{\mu}_i), \tag{36}$$

where $\pi_i = \Pr_P(y = i)$ and $\boldsymbol{\mu}_i = \mathbb{E}_P[\bar{\mathbf{x}} \mid y = i]$.

Similarly, the expectation under distribution $Q$ is given by

$$\mathbb{E}_Q[\phi(\bar{\mathbf{x}}, y)] = \sum_{i=1}^{C} \pi_i'\, (\mathbf{e}_i \otimes \boldsymbol{\mu}_i'), \tag{37}$$

where $\pi_i' = \Pr_Q(y = i)$ and $\boldsymbol{\mu}_i' = \mathbb{E}_Q[\bar{\mathbf{x}} \mid y = i]$.

Substituting the above expressions into the MMD definition yields

$$\mathrm{MMD}_k^2(P, Q) = \left\| \sum_{i=1}^{C} \left( \pi_i\, \mathbf{e}_i \otimes \boldsymbol{\mu}_i - \pi_i'\, \mathbf{e}_i \otimes \boldsymbol{\mu}_i' \right) \right\|_2^2. \tag{38}$$

Note that the subspaces $\{\mathbf{e}_i \otimes \mathbb{R}^d\}_{i=1}^{C}$ are mutually orthogonal. Moreover, $\pi_i' = \pi_i$ for all $i \in \{1, \ldots, C\}$, because the condensed data follows the same label distribution as the original data. Therefore, the MMD loss can be written as

$$\mathrm{MMD}_k^2(P, Q) = \sum_{i=1}^{C} \|\pi_i \boldsymbol{\mu}_i - \pi_i' \boldsymbol{\mu}_i'\|_2^2 = \sum_{i=1}^{C} \pi_i^2 \|\boldsymbol{\mu}_i - \boldsymbol{\mu}_i'\|_2^2.. \tag{39}$$

For any unit vectors $a$ and $b$ (i.e., $\|a\| = \|b\| = 1$), the following identity holds:

$$\|a - b\|^2 = \|a\|^2 + \|b\|^2 - 2a^\top b = 2 - 2a^\top b. \tag{40}$$

Let $a = \mathbf{C}_i/\|\mathbf{C}_i\|$, $b = u_i = \mathbf{C}_i'/\|\mathbf{C}_i'\|$, according to Eq. (40), we have

$$1 - \frac{\mathbf{C}_i^\top \cdot \mathbf{C}_i'}{\|\mathbf{C}_i\| \|\mathbf{C}_i'\|} = \frac{1}{2} \|\frac{\mathbf{C}_i}{\|\mathbf{C}_i\|} - \frac{\mathbf{C}_i'}{\|\mathbf{C}_i'\|}\|^2 \tag{41}$$

Thus, the alignment term $\sum_{i=1}^{C} \left( 1 - \frac{\mathbf{C}_i^\top \cdot \mathbf{C}_i'}{\|\mathbf{C}_i\| \|\mathbf{C}_i'\|} \right) = \sum_{i=1}^{C} \frac{1}{2} \|\frac{\mathbf{C}_i}{\|\mathbf{C}_i\|} - \frac{\mathbf{C}_i'}{\|\mathbf{C}_i'\|}\|^2$.

Recall that the class prototype $\mathbf{C}_i$ is obtained by aggregating the normalized embeddings of samples from class $i$. Thus, up to a scaling factor, $\mathbf{C}_i$ provides an empirical estimate of the class-conditional mean $\boldsymbol{\mu}_i = \mathbb{E}[\bar{\mathbf{x}} \mid y = i]$. After normalization, the prototype directions align with the corresponding $\boldsymbol{\mu}_i$ for both the original and condensed data, i.e., $\frac{\mathbf{C}_i}{\|\mathbf{C}_i\|} \propto \boldsymbol{\mu}_i$, $\frac{\mathbf{C}_i'}{\|\mathbf{C}_i'\|} \propto \boldsymbol{\mu}_i'$.

Therefore, minimizing the alignment loss is equivalent to minimizing the $\ell_2$ distance between normalized class-conditional means, which acts as a surrogate for class-wise MMD. This completes the proof.

$$\square$$

### E.4. Proof of Proposition 3.5

**Proposition E.4.** *Assume the total positive cross-class similarity is bounded by $\sum_{i \neq j}[u_i^\top u_j']_+ \leq \varepsilon$, where $[x]_+ = \max(x, 0)$. Then the average class-level margin satisfies*

$$\frac{1}{C}\sum_{i=1}^{C} m_i \; \geq \; \frac{1}{C}\sum_{i=1}^{C} u_i^\top u_i' \; - \; \frac{\varepsilon}{C}. \tag{42}$$

*Proof.* For any set of real numbers $\{a_j\}_{j \neq i}$, we have the following inequality:

$$\max_{j \neq i} a_j \; \leq \; \sum_{j \neq i}[a_j]_+, \tag{43}$$

where $[x]_+ = \max(x, 0)$. Applying this inequality to $a_j = u_i^\top u_j'$ yields

$$\max_{j \neq i} u_i^\top u_j' \; \leq \; \sum_{j \neq i}[u_i^\top u_j']_+. \tag{44}$$

Therefore,

$$m_i = u_i^\top u_i' - \max_{j \neq i} u_i^\top u_j' \; \geq \; u_i^\top u_i' - \sum_{j \neq i}[u_i^\top u_j']_+. \tag{45}$$

Averaging the above inequality over $i = 1, \ldots, C$, we obtain

$$\frac{1}{C}\sum_{i=1}^{C} m_i \; \geq \; \frac{1}{C}\sum_{i=1}^{C} u_i^\top u_i' - \frac{1}{C}\sum_{i=1}^{C}\sum_{j \neq i}[u_i^\top u_j']_+. \tag{46}$$

Noting that $\sum_{i=1}^{C}\sum_{j \neq i}[u_i^\top u_j']_+ = \sum_{i \neq j}[u_i^\top u_j']_+$, and using the assumption $\sum_{i \neq j}[u_i^\top u_j']_+ \leq \varepsilon$, we have

$$\frac{1}{C}\sum_{i=1}^{C} m_i \; \geq \; \frac{1}{C}\sum_{i=1}^{C} u_i^\top u_i' - \frac{\varepsilon}{C}. \tag{47}$$

This completes the proof. $\qquad\square$

### E.5. Proof of Proposition 3.8

**Proposition E.5.** *Given the node-wise fine-grained discrimination loss $\ell_i$, the probability of the Mis-ranking Event $\mathcal{E}_i$ is upper-bounded by:*

$$\Pr(\mathcal{E}_i) \; \leq \; \mathbb{E}\big[e^{\ell_i} - 1\big]. \tag{48}$$

*Proof.* If the Mis-ranking Event $\mathcal{E}_i$ occurs, then there exists a negative sample $q \in Q(i)$ such that $s_{i,q} \geq s_{i,p}$. This implies $\exp(s_{i,q} - s_{i,p}) \geq 1$, and consequently,

$$\sum_{q \in Q(i)} \exp(s_{i,q} - s_{i,p}) \geq 1. \tag{49}$$

Therefore, the event $\mathcal{E}_{i,p}$ is contained in the event $\left\{\sum_{q \in Q(i)} \exp(s_{i,q} - s_{i,p}) \geq 1\right\}$, which yields

$$\Pr(\mathcal{E}_{i,p}) \leq \Pr\left(\sum_{q \in Q(i)} \exp(s_{i,q} - s_{i,p}) \geq 1\right). \tag{50}$$

Let $X = \sum_{q \in Q(i)} \exp(s_{i,q} - s_{i,p})$, which is a nonnegative random variable. By Markov's inequality, we have

$$\Pr(X \geq 1) \leq \mathbb{E}[X], \tag{51}$$

and thus

$$\Pr(\mathcal{E}_i) \leq \mathbb{E}\left[\sum_{q \in Q(i)} \exp(s_{i,q} - s_{i,p})\right]. \tag{52}$$

From the definition of the fine-grained discrimination loss $\ell_i$, we obtain

$$e^{\ell_i} = \frac{\exp(s_{i,p}) + \sum_{q \in Q(i)} \exp(s_{i,q})}{\exp(s_{i,p})} = 1 + \sum_{q \in Q(i)} \exp(s_{i,q} - s_{i,p}), \tag{53}$$

which implies

$$\sum_{q \in Q(i)} \exp(s_{i,q} - s_{i,p}) = e^{\ell_i} - 1. \tag{54}$$

Substituting this relation into the inequality (52) completes the proof. $\square$

