# OpenReview forum: "Anchor-guided Hypergraph Condensation with Dual-level Discrimination"
_ICML.cc/2026/Conference — ICML 2026 regular_

### Official Review · Reviewer_T4m6 · 2026-03-12

**Soundness:** 2
**Presentation:** 4
**Significance:** 3
**Originality:** 4
**Overall Recommendation:** 4
**Confidence:** 3

**Summary:**

The paper proposes Anchor-guided Hypergraph Condensation with Dual-level Discrimination (AHGCDD), introducing a new hypergraph condensation pipeline.

**Compliance With Llm Reviewing Policy:**

Affirmed.

**Final Justification:**

The authors provided good additional evaluation, which makes me change my score from 3 to 4.

**Key Questions For Authors:**

Could you evaluate AHGCDD at compression ratios comparable to HG-Cond (e.g., 5% or 7%) to directly assess effectiveness?

HG-Cond reports around 88% accuracy on PubMed at a 7% reduction ratio. How does AHGCDD perform under similar conditions?

Why were datasets like CiteSeer and 20 Newsgroups not included, given their use in prior work?

**Limitations:**

I am concerned that the model could have limitations that the authors hide by leaving some evaluations out (details in the above sections).

**Strengths And Weaknesses:**

**Soundness**
The method appears technically reasonable and is well supported by theoretical results.

However, the evaluation raises several questions. The authors conclude:

“Extensive experiments on six real-world benchmarks verify the superiority of AHGCDD in terms of effectiveness and efficiency.”

The method is mainly assessed at stronger compression ratios than the baselines. While this highlights efficiency, it weakens claims of superior effectiveness. For instance, HG-Cond reports higher accuracy on PubMed at a 7% reduction ratio than the best result reported here. Could the authors evaluate AHGCDD at comparable ratios (e.g., 5% or 7%)? Similarly, all other datasets are compressed more aggressively than in prior work, making direct comparison difficult.

The benchmark selection also raises questions. Closely related work, including HG-Cond, evaluates additional datasets such as CiteSeer and 20News, which are not included. Including these datasets would strengthen the evaluation.

**Presentation**
The paper is clearly written and well-structured. The methodological description is understandable, and the theoretical analysis helps motivate design choices.

**Significance**
Hypergraph condensation is a young but relevant research direction. The authors show that AHGCDD outperforms prior work, but some questions remain regarding certain experimental settings. Even if these concerns are valid and the method does not consistently beat existing baselines, achieving strong compression with substantial efficiency improvements still represents a valuable contribution.

**Originality**
The proposed pipeline appears novel and differs substantially from existing approaches for the task.

Generally, the method is promising, but the current evaluation does not convincingly demonstrate superior effectiveness. If the authors provide results at comparable compression ratios or revise the conclusions to more accurately reflect the findings, I would consider increasing my score to accept.

---

> ### Author Rebuttal · Authors · 2026-03-31
>
> We sincerely thank the reviewer for the thoughtful comments. Our responses are provided below.
>
> > **Q1：Evaluate AHGCDD at compression ratios comparable to HG-Cond.**
>
> **A:** Thank you for the insightful question. We agree that evaluation under comparable ratios on the same datasets is essential for a fair assessment. To this end, **we follow the reviewer’s suggestion and conduct additional experiments under comparable condensation ratios.** Specifically, we evaluate AHGCDD on five datasets (Cora, Cora-CA, PubMed, CiteSeer, and 20News), including CiteSeer and 20News from the HG-Cond benchmark. We adopt the same settings as HG-Cond, using a 5% condensation ratio on 20News and 7% on the remaining datasets. For evaluation, we use HGNN [1] (Table A1), a representative HNN architecture, and additionally report results using AllDeepSets [2] (Table A2), as used in HG-Cond, to ensure a fair and consistent comparison.
>
> The results in Table A1 and Table A2 show that **under comparable condensation ratios, AHGCDD outperforms HG-Cond on 4 out of 5 datasets** under both HGNN and AllDeepSets evaluation models, indicating strong effectiveness at larger ratios.
>
> **Table A1：Evaluation on Comparable Ratio (HGNN)**
>
> |Method|Cora|Cora-CA|PubMed|CiteSeer|20News|
> |-|-|-|-|-|-|
> |HG-Cond|77.55±1.09|80.42±2.06|**82.29±0.32**|71.14±0.83|77.29±1.15|
> |AHGCDD|**78.10±1.51**|**80.80±1.61**|81.93±0.48|**71.98±0.97**|**77.51±0.44**|
> |Whole|77.90±1.17|82.84±0.46|86.17±0.52|72.21±0.69|80.75±0.43|
>
> **Table A2：Evaluation on Comparable Ratio (AllDeepSets)**
>
> |Method|Cora|Cora-CA|PubMed|CiteSeer|20News|
> |-|-|-|-|-|-|
> |HG-Cond|73.86±2.33|76.37±2.11|**87.79±1.65**|69.69±1.12|80.33±1.04|
> |AHGCDD|**75.45±2.68**|**76.81±1.99**|87.22±0.78|**72.71±0.79**|**80.84±0.57**|
> |Whole|76.26±0.96|78.47±1.17|87.94±0.15|71.93±0.91|81.01±0.33|
>
> In our main experiments, we adopt lower condensation ratios to **better reflect practical scenarios and align with the goal of data condensation,** i.e., preserving training utility with as little data as possible. Importantly, maintaining strong performance under such low ratios is more challenging, further demonstrating the effectiveness and robustness of our method.
>
> **We thank the reviewer for this helpful suggestion and will include the above results in the final version.**
>
> > **Q2： How does AHGCDD perform on PubMed at $r=7$%?**
>
> **A:** Thank you for the question. Under the same setting as HG-Cond (7% condensation ratio with AllDeepSets as the evaluation model), AHGCDD achieves 87.22 accuracy on PubMed, compared to 87.79 from HG-Cond (Table A2). Notably, both results are close to the full-data performance (87.94), indicating strong performance preservation under this ratio.
>
> > **Q3：Why were datasets like CiteSeer and 20 Newsgroups not included?**
>
> **A:** Thank you for the question. Following the reviewer’s suggestion, **we have included results on CiteSeer and 20 Newsgroups in Q1** for direct comparison with prior work. As shown in Table A1 and Table A2, AHGCDD achieves better performance than HG-Cond under comparable condensation ratios (e.g., 5% or 7%) on both datasets. The reason these two datasets are not included in our main experiments is that **we prioritize a diverse range of dataset scales** to better assess efficiency gains and generalization. As **CiteSeer and 20 Newsgroups are similar in scale to Cora and PubMed**, we instead incorporate larger-scale datasets (e.g., Yelp with over 600K hyperedges and Trivago with over 170K nodes). This better reflects real-world scenarios where hypergraphs are typically large, aligning with the practical motivation of data condensation.
>
> ---
>
> **References**
>
> [1] Hypergraph neural networks. AAAI 2019
>
> [2] You are AllSet: A Multiset Function Framework for Hypergraph Neural Networks. ICLR 2022

---

> > ### Author Rebuttal · Reviewer_T4m6 · 2026-04-03
> >
> > Thank you for addressing my concerns. I raise the score, given the additional evaluations.

---

> > > ### Author Response · Authors · 2026-04-05
> > >
> > > Thank you so much for raising your score to recommend acceptance! We truly appreciate your engagement in the discussion and are glad that our responses have addressed your concerns.
> > >
> > > We will incorporate the additional evaluations suggested into the final version to further strengthen the paper.
> > >
> > > Thank you again for your time and attention.

---

### Official Review · Reviewer_QgcB · 2026-03-12

**Soundness:** 3
**Presentation:** 3
**Significance:** 3
**Originality:** 3
**Overall Recommendation:** 4
**Confidence:** 4

**Summary:**

This paper proposes AHGCDD for compressing a large attributed hypergraph into a much smaller synthetic hypergraph while preserving its discriminative capability. The method first employs Heat Kernel PageRank (HKPR) to obtain structure-aware node representations, and then generates synthetic hyperedges through an anchor-guided mechanism, enabling joint optimization of features and structures. In addition, a dual-level discrimination objective is introduced to align the original and condensed graphs at both class-level and sample-level representations. Experiments on several real-world hypergraph datasets demonstrate that the proposed method achieves superior performance compared with existing approaches while significantly reducing computational and memory costs.

**Compliance With Llm Reviewing Policy:**

Affirmed.

**Key Questions For Authors:**

W1: Over-reliance on feature similarity for hyperedge generation. The anchor-guided hyperedge generation mechanism constructs hyperedges by feeding the concatenated feature vectors of anchor and candidate nodes into an MLP to compute similarity scores. This design inherently confines hyperedge formation to the feature similarity space. However, higher-order interactions in real-world hypergraphs—such as author groups in co-authorship networks or product bundles in co-purchase networks—often arise from structural patterns, semantic relationships, or actual interaction behaviors rather than purely feature-based similarities. As a result, this approach may fail to capture the genuine higher-order structural regularities present in complex real-world data.

W2: Beyond feature similarity, real-world hypergraphs often encode rich structural patterns and semantic relationships that are crucial for accurately modeling higher-order interactions. For instance, motif-based connections in social networks or topic consistency among documents provide essential information that cannot be captured by feature similarity alone. The current framework does not incorporate such information into the hyperedge generation process, potentially limiting its ability to reflect the true underlying interaction structures found in complex real-world data.

W3: Absence of evaluation on heterophilic hypergraphs. Many real-world hypergraphs exhibit heterophily, where nodes from different classes are frequently connected. Given that the proposed hyperedge generation mechanism is primarily driven by feature similarity, it is inherently biased toward homophilic structures. The lack of experiments on heterophilic hypergraphs leaves the generalizability of the method unclear, as its performance under such conditions remains unexamined and unvalidated.

**Limitations:**

Yes.

**Strengths And Weaknesses:**

Strength
S1: The paper introduces a novel framework for hypergraph condensation by employing an anchor-guided hyperedge generation mechanism, which jointly learns synthetic structures and node features, avoiding the mismatch issue between structure and feature optimization in previous approaches.
S2: Compared with prior methods relying on expensive training trajectory matching, the proposed approach employs discriminative objectives to avoid repeated model training, significantly reducing both computational time and memory consumption.
S3: The proposed method is evaluated on multiple real-world hypergraph datasets with comparisons against several baselines, demonstrating superior performance and efficiency.

Weaknesses
W1: Over-reliance on feature similarity for hyperedge generation. The anchor-guided hyperedge generation mechanism constructs hyperedges by feeding the concatenated feature vectors of anchor and candidate nodes into an MLP to compute similarity scores. This design inherently confines hyperedge formation to the feature similarity space. However, higher-order interactions in real-world hypergraphs—such as author groups in co-authorship networks or product bundles in co-purchase networks—often arise from structural patterns, semantic relationships, or actual interaction behaviors rather than purely feature-based similarities. As a result, this approach may fail to capture the genuine higher-order structural regularities present in complex real-world data.

W2: Beyond feature similarity, real-world hypergraphs often encode rich structural patterns and semantic relationships that are crucial for accurately modeling higher-order interactions. For instance, motif-based connections in social networks or topic consistency among documents provide essential information that cannot be captured by feature similarity alone. The current framework does not incorporate such information into the hyperedge generation process, potentially limiting its ability to reflect the true underlying interaction structures found in complex real-world data.

W3: Absence of evaluation on heterophilic hypergraphs. Many real-world hypergraphs exhibit heterophily, where nodes from different classes are frequently connected. Given that the proposed hyperedge generation mechanism is primarily driven by feature similarity, it is inherently biased toward homophilic structures. The lack of experiments on heterophilic hypergraphs leaves the generalizability of the method unclear, as its performance under such conditions remains unexamined and unvalidated.

---

> ### Author Rebuttal · Authors · 2026-03-31
>
> We sincerely thank the reviewer for the thoughtful comments. Our responses are provided below.
>
> > **W1：Over-reliance on feature similarity for hyperedge generation.**
>
> **A:** Thank you for your insightful comments. We agree that modeling hyperedges solely based on feature similarity may be insufficient to capture the complex higher-order interactions in real-world hypergraphs. However, we would like to clarify that **our method goes beyond such similarity-based modeling.** Specifically, the condensed node features $\mathbf{X}'$ are initialized via HKPR diffusion, which **adaptively integrates high-order structural information** from local to global neighborhoods with feature semantics, yielding enriched node representations. During hyperedge generation, we compute the association strength between each candidate node $v_{j}^{\prime}$ and the anchor-induced hyperedge $e_{i}^{\prime}$ via an MLP, where the ordered input $[\mathbf{X}_i', \mathbf{X}_j']$ encodes their distinct roles, with the anchor providing hyperedge context and the candidate providing node-specific information. **Importantly, this hyperedge generator is learnable and task-driven, being optimized by our dual-level discrimination objective rather than any predefined similarity**. The coarse-grained objective aligns class-wise distributions while preserving global separability, whereas the fine-grained objective further preserves local discriminative geometry. As a result, our method captures meaningful higher-order structural and semantic relationships beyond feature proximity, making it suitable for complex real-world hypergraphs.
>
> > **W2： Limited modeling of rich structural and semantic relationships.**
>
> **A:** Thank you for your constructive feedback. We agree that real-world hypergraphs often involve rich structural patterns and semantic relationships beyond feature similarity. **In this regard, we would like to clarify that our method is already designed to capture such information.** Specifically, the HKPR-based diffusion integrates high-order structural information from local to global neighborhoods with feature semantics, which can be interpreted as a spectral low-pass filtering process that preserves smooth structural patterns (Theorem 3.1). Moreover, during optimization, we introduce a dual-level discrimination objective built upon HKPR-based high-order representations of both the original and condensed hypergraphs. This objective drives the learning of the condensed data, aligning global class-wise distributions while preserving local discriminative geometry. In this process, rich information from the original graph, including complex interaction patterns and task-relevant higher-order relationships, is effectively transferred to the condensed data. The theoretical justification of this objective is provided in Sec. 3.3.
>
> > **W3：Absence of evaluation on heterophilic hypergraphs.**
>
> **A:**  Thank you for the valuable comments. We agree that evaluating on heterophilic hypergraphs is important for assessing generalizability. **Our method does not inherently favor homophilic structures**, as hyperedge generation is not restricted to feature similarity (clarified in Q1). Instead, it leverages a learnable function over high-order node representations, optimized via a task-aware discrimination objective, which avoids bias toward homophily.
>
> Following the reviewer’s suggestion, **we further include experiments on three heterophilic datasets** (Actor, Amazon-ratings, and Pokec [1]) under a 1% condensation ratio. As shown in Table A1, our method consistently achieves state-of-the-art performance and preserves over 90% of the full-data performance across all datasets, demonstrating strong robustness and generalizability in heterophilic settings.
>
> **Table A1：Evaluation on Heterophilic Datasets.**
>
> |Dataset|Herding|HGCPA|HG-Cond|AHGCDD|Whole|
> |-|-|-|-|-|-|
> |Actor|70.28±1.06|66.40±3.01|71.64±0.57|**72.29±1.21**|77.83±0.37|
> |Amazon-ratings|24.16±0.31|23.12±1.86|24.53±0.77|**25.41±0.39**|28.05±0.28|
> |Pokec|53.03±0.98|52.37±0.76|53.55±0.21|**55.04±0.47**|57.87±0.76|
>
> **We thank the reviewer for this helpful suggestion and will include these additional results in the final version.**
>
> ---
>
> **References**
>
> [1] When hypergraph meets heterophily: New benchmark datasets and baseline. AAAI 2025

---

> > ### Author Rebuttal · Reviewer_QgcB · 2026-04-04
> >
> > The authors have provided a detailed and thoughtful response that satisfactorily addresses the majority of my concerns. Accordingly, I will retain my original score.

---

> > > ### Author Response · Authors · 2026-04-05
> > >
> > > Thank you for your thoughtful and positive feedback. We sincerely appreciate your careful evaluation and are glad that our responses have addressed your concerns.
> > >
> > > We will ensure that the additional results are incorporated into the final version.
> > >
> > > Thank you again for your time and support.

---

### Official Review · Reviewer_ZAvW · 2026-03-14

**Soundness:** 3
**Presentation:** 3
**Significance:** 4
**Originality:** 4
**Overall Recommendation:** 4
**Confidence:** 3

**Summary:**

This paper studies hypergraph condensation, aiming to compress a large hypergraph into a much smaller synthetic one while preserving node-classification utility. It argues that prior methods use a decoupled training pipeline, where the structure generator is pre-trained and not jointly optimized with condensed features, causing possible structure-feature misalignment and high optimization cost. To address this, the paper proposes AHGCDD, which includes three components: HKPR-based node initialization, anchor-guided hyperedge generation, and a dual-level discrimination objective combining coarse-grained class alignment with fine-grained instance discrimination. The paper also provides theoretical support for the objective and reports strong results on six real-world benchmarks.

**Compliance With Llm Reviewing Policy:**

Affirmed.

**Key Questions For Authors:**

1. The current experiments focus on node classification. How well does the condensed hypergraph transfer across different HNN backbones or related downstream tasks?
2. Can the authors clarify more explicitly which parts of the efficiency gain come from removing repeated HNN retraining versus from the anchor-guided structure generation design itself?
3. The HKPR diffusion for the original graph is computed once and fixed. Was any ablation conducted to compare this against a learnable or dynamically updated diffusion during condensation?

**Limitations:**

Yes

**Strengths And Weaknesses:**

Strengths
- The paper identifies a concrete technical limitation in prior hypergraph condensation methods, namely the decoupled optimization of synthetic features and synthetic structure. This diagnosis is technically meaningful because structure generation directly affects the usefulness of condensed data in hypergraph settings.
- The anchor-guided hyperedge generation module is a strong part of the method. It gives a clear mechanism for tying structure synthesis to the current synthetic features, which is more reasonable than relying on a separately pre-trained and fixed structure generator.
- The empirical section is also strong overall. The paper evaluates on six benchmarks with standard train, validation, and test splits, and compares against both traditional coreset methods and hypergraph-condensation baselines, including HG-Cond, HG-Cond-NHL, and a hypergraph-adapted GCPA baseline.

Weaknesses
+ Although the method is technically well structured, it also introduces several interacting components at once, including HKPR initialization, anchor-guided structure synthesis, adaptive thresholds, coarse-grained discrimination, fine-grained discrimination, and dynamic weighting. This makes it somewhat difficult to isolate which component is most responsible for the improvement beyond the ablation trends.
+ My main concern is that the evaluation is still centered primarily on node classification. While the appendix includes cross-architecture transfer results on multiple HNN backbones, broader validation on more downstream settings would further strengthen the paper’s generality claims.
+ The dynamic weighting schedule lacks sufficient ablation. The cosine annealing schedule in Eq. (13) is parameter-free and intuitive, but Table 4 only compares it against static equal weighting—the weakest possible baseline. It remains unclear whether the specific cosine schedule is critical, or whether any monotonic coarse-to-fine reweighting would yield similar gains.

---

> ### Author Rebuttal · Authors · 2026-03-31
>
> We sincerely thank the reviewer for the thoughtful comments. Our responses are provided below.
>
> > **W1：Hard to identify the top contributing component**
>
> **A:** Thank you for your comments. First, our ablation studies demonstrate the contribution of each component. As shown in Fig. 3, HKPR-based initialization consistently outperforms standard hypergraph propagation and no propagation. In the structure generation stage, anchor-guided generation with adaptive sparsity achieves better performance than fixed thresholds. In the optimization stage, both coarse- and fine-grained discrimination contribute to performance, and dynamic weighting further improves over static weighting (Table 4). At the same time, these components are interdependent. Specifically, HKPR-based initialization provides structure-aware representations that support both anchor-guided generation and discrimination learning. Anchor-guided generation is optimized under the dual-level objective, while the discrimination objective is defined over HKPR-based representations of both original and condensed data. As a result, they are not designed to function independently, and the improvement arises from their joint effect.
>
> > **W2 & Q1：Broader downstrem settings.**
>
> **A:** Thank you for your comments. For cross-backbone transferability, we have already provided evaluation results in Appendix D.1 (also noted in W2).
> For broader downstream settings, we agree that they would further strengthen generality, and have provided experiments on node clustering (Table A1) and node retrieval (Table A2) [1], measured by NMI (Normalized Mutual Information) and MAP (Mean Average Precision), respectively.
> The results show that AHGCDD consistently achieves SOTA performance and retains over 85% of the full-data performance on both tasks, demonstrating its strong generalization capability.
>
> **Table A1: Node Clustering**
>
> |Dataset|Herding|HG-Cond|AHGCDD|Whole|
> |-|-|-|-|-|
> |Cora (r=1\%)|23.27±3.25|43.75±1.86|**47.58±1.84**|50.46±2.55|
> |DBLP-CA (r=0.5\%)|63.23±0.62|70.46±1.70|**71.99±0.67**|75.10±0.50|
> |MAG-PM (r=0.25\%)|18.53±0.59|24.56±0.16|**26.95±0.32**|31.48±0.75|
>
> **Table A2: Node Retrieval**
>
> |Dataset|Herding|HG-Cond|AHGCDD|Whole|
> |-|-|-|-|-|
> |Cora (r=1\%)|38.18±1.36|67.13±0.70|**70.95±0.64**|79.06±0.65|
> |DBLP-CA (r=0.5\%)|73.64±0.52|79.66±0.84|**81.65±0.75**|83.79±0.21|
> |MAG-PM (r=0.25\%)|24.48±0.68|28.44±0.68|**31.42±0.95**|33.88±0.50|
>
> > **W3：The dynamic weighting lacks sufficient ablation.**
>
> **A:** Thank you for the suggestion. We additionally evaluate two monotonic coarse-to-fine schedules, **Linear** and **Step**, to extend the ablation on dynamic weighting.
>
> - **Linear:** $ w_c(t)=1-\frac{t}{T}, \; w_f(t)=\frac{t}{T} $
> - **Step:** $ w_c(t)=1 \; (t<\tfrac{T}{2}), \; 0 \; (t\ge\tfrac{T}{2}), \; w_f(t)=1-w_c(t) $
>
> As shown in Table A3, monotonic reweighting generally outperforms static weighting, supporting the effectiveness of our coarse-to-fine design. Linear and Cosine perform best, while Step is less stable, likely due to the abrupt transition at $t=\tfrac{T}{2} $, indicating the importance of smooth scheduling. Cosine further improves over Linear, possibly because its slower early-stage decay of $w_c(t)$, which better preserves global separability.
>
> **Table A3: Reweighting Schedule**
>
> |Dataset|Static|Linear|Step|Cosine|
> |-|-|-|-|-|
> |Cora|75.63±1.42|75.92±1.18|75.33±1.34|76.48±1.58|
> |DBLP-CA|87.79±0.55|88.36±0.52|87.97±0.47|88.80±0.65|
> |MAG-PM|56.84±0.81|57.22±0.80|56.43±2.48|57.55±0.54|
>
> > **Q2：Source of efficiency gain**
>
> **A:** Thank you for the question. We clarify that the main efficiency bottleneck in prior HG-Cond lies in its gradient- and parameter-matching objectives, which **require repeated HNN retraining** to store multiple trajectories. Our dual-level discrimination objective **removes this need** while preserving training utility, leading to significant efficiency gains. Meanwhile, anchor-guided generation mainly enables task-aware joint optimization of structure and features, rather than contributing to efficiency.
>
> > **Q3：Ablation on learnable and dynamic diffusion**
>
> **A:** Thank you for the question. We understand the concern that learnable or dynamically updated diffusion may further improve performance. However, such designs typically require pretraining or repeated inference on the original large-scale graph during condensation, leading to substantial computational overhead. Moreover, since the original graph remains fixed, dynamically updating the diffusion may introduce instability to the discrimination signal. In contrast, our HKPR diffusion is computed once to provide efficient, structure-aware node initialization, and is theoretically grounded (Theorem 3.1). Therefore, while potentially beneficial, such designs may not be well aligned with our design principles.
>
> ---
>
> **References**
>
> [1] VilLain: Self-supervised learning on homogeneous hypergraphs without features via virtual label propagation. ACM Web Conference 2024

---

> > ### Author Rebuttal · Reviewer_ZAvW · 2026-04-02
> >
> > I thank the authors for their detailed and thoughtful rebuttal, which effectively addresses most of my concerns and provides valuable clarifications. I will maintain my score.

---

> > > ### Author Response · Authors · 2026-04-05
> > >
> > > We sincerely appreciate your recognition and support of our work. We are glad that our responses have addressed your concerns.
> > >
> > > We will ensure that the additional evaluations and clarifications are incorporated into the final version.
> > >
> > > Thank you again for your time and valuable feedback.

---

### Decision · Program_Chairs · 2026-04-30

**Decision:**

Accept (regular)

**Comment:**

This paper studies hypergraph condensation and proposes AHGCDD, a technically strong and efficient framework with joint structure-feature optimization and consistently strong empirical performance. The reviewers generally agreed that the paper identifies a meaningful limitation of prior hypergraph condensation methods and offers a well-motivated solution, and the rebuttal resolved several of the main concerns raised in the initial reviews. In particular, the authors clarified the source of the efficiency gain, added broader evaluations including comparable compression-ratio experiments and additional downstream tasks, extended the ablation on the dynamic weighting schedule, and provided new results on heterophilic datasets.

The remaining concerns are relatively limited and mostly concern scope and interpretation rather than correctness. In particular, one reviewer noted that the anchor-guided hyperedge generation is still heavily feature-driven, even though the rebuttal clarified that high-order structural information is incorporated through HKPR-based initialization and task-driven optimization. More broadly, while the added heterophilic experiments strengthen the paper, the generality of the approach beyond the studied settings remains a reasonable point for future discussion.

Overall, the paper received consistently positive reviewer feedback after rebuttal, with the main issues largely addressed. I recommend acceptance of the paper as a poster.